# Oxytocin improves behavioral and electrophysiological deficits in a novel Shank3-deficient rat

Hala Harony-Nicolas[1,2], Maya Kay[3†], Johann du Hoffmann[1,4†], Matthew E Klein[5], Ozlem Bozdagi-Gunal[1,2], Mohammed Riad[1,2], Nikolaos P Daskalakis[2], Sankalp Sonar[1,2], Pablo E Castillo[5], Patrick R Hof[1,4,6], Matthew L Shapiro[4,6], Mark G Baxter[4,6], Shlomo Wagner[3], Joseph D Buxbaum[1,2,4,6,7,8]*

[1]Seaver Autism Center for Research and Treatment, Icahn School of Medicine at Mount Sinai, New York, United States; [2]Department of Psychiatry, Icahn School of Medicine at Mount Sinai, New York, United States; [3]Sagol Department of Neurobiology, University of Haifa, Haifa, Israel; [4]Friedman Brain Institute, Icahn School of Medicine at Mount Sinai, New York, United States; [5]Dominick P. Purpura Department of Neuroscience, Albert Einstein College of Medicine, New York, United States; [6]Fishberg Department of Neuroscience, Icahn School of Medicine at Mount Sinai, New York, United States; [7]Department of Genetics and Genomic Sciences, Icahn School of Medicine at Mount Sinai, New York, United States; [8]The Mindich Child Health and Development Institute, Icahn School of Medicine at Mount Sinai, New York, United States

*For correspondence: joseph.
buxbaum@mssm.edu

†These authors contributed
equally to this work

Competing interests: The
authors declare that no
competing interests exist.

Reviewing editor: Peggy
Mason, University of Chicago,
United States

**Abstract** Mutations in the synaptic gene *SHANK3* lead to a neurodevelopmental disorder known as Phelan-McDermid syndrome (PMS). PMS is a relatively common monogenic and highly penetrant cause of autism spectrum disorder (ASD) and intellectual disability (ID), and frequently presents with attention deficits. The underlying neurobiology of PMS is not fully known and pharmacological treatments for core symptoms do not exist. Here, we report the production and characterization of a *Shank3*-deficient rat model of PMS, with a genetic alteration similar to a human SHANK3 mutation. We show that *Shank3*-deficient rats exhibit impaired long-term social recognition memory and attention, and reduced synaptic plasticity in the hippocampal-medial prefrontal cortex pathway. These deficits were attenuated with oxytocin treatment. The effect of oxytocin on reversing non-social attention deficits is a particularly novel finding, and the results implicate an oxytocinergic contribution in this genetically defined subtype of ASD and ID, suggesting an individualized therapeutic approach for PMS.

## Introduction

Phelan McDermid syndrome (PMS) is a neurodevelopmental disorder characterized by intellectual disability (ID), absent or delayed speech, neonatal hypotonia, attention deficits and autism spectrum disorder (ASD). The neurobehavioral manifestations of PMS are caused by heterozygous mutations/deletions in the *SHANK3* gene leading to a reduced expression of the SHANK3 protein. Shank3 is a key structural component of the glutamatergic postsynaptic density (PSD), and interacts with glutamate receptors and cytoskeletal elements to regulate glutamate signaling and synaptic plasticity (*Kreienkamp, 2008*). It has been estimated that more than 80% of individuals with PMS meet ASD diagnostic criteria and that ~0.5–1% of ASD cases, 1–2% of ID cases, and up to 2% of cases with

**eLife digest** Phelan-McDermid syndrome is a genetic disorder on the autism spectrum that affects how children develop in several ways, with additional symptoms including attention deficits, delays in learning to speak and motor problems. This syndrome is known to be caused by changes in a single gene known as SHANK3 that disrupt communication between brain cells involved in memory and learning. However, we do not know how these changes relate to the symptoms of Phelan-McDermid syndrome.

To understand how genetic changes affect the human brain, researchers often carry out experiments in rats or other small rodents because they have brains that are similar to ours. Harony-Nicolas et al. genetically modified rats to carry changes in the SHANK3 gene that reflect those found in people with Phelan-McDermid syndrome. The rats had disabilities related to those seen in Phelan-McDermid syndrome, including limits in long-term social memory and reduced attention span. They also showed changes in the connections between important parts of the brain. Therefore, studying these rats could help us to understand the link between molecular and cellular changes in the brain and how they affect people with Phelan-McDermid syndrome, and associated symptoms.

Previous studies have shown that a chemical called oxytocin, which is naturally produced by the brain, helps to form bonds between individuals and can cause positive feelings in relation to certain memories. Harony-Nicolas et al. found treating the rats with oxytocin boosted social memory and led to improvements in other symptoms of Phelan-McDermid syndrome. In particular, oxytocin treatment helped to increase the attention span of the rats.

Rats with changes in the SHANK3 gene will be a useful tool for future research into Phelan-McDermid syndrome, particularly in understanding how it affects the connections between brain cells, leading to the symptoms of Phelan-McDermid syndrome. A future challenge will be to find out whether oxytocin has the potential to be developed into a therapy to treat Phelan-McDermid syndrome in humans. Since there is evidence that SHANK3 is involved in other forms of autism, these rats will also be useful in understanding the other ways in which autism can develop.

both ASD and ID harbor a *SHANK3* mutation, which makes it one of the more common single locus causes of ASD and ID (*Gong et al., 2012*; *Soorya et al., 2013*; *Leblond et al., 2014*). Despite its prevalence, PMS is less well studied than other single locus genetic disorders such as Fragile X or Rett syndromes. To date, no pharmaceutical compounds targeting core symptoms of PMS are available. To address this lack of effective therapeutics, several mouse lines with distinct *Shank3* gene mutations have been developed to help understand the neurobiology of the syndrome and as a means of ultimately developing and assessing potential therapeutics (*Bozdagi et al., 2010*; *Peça et al., 2011*; *Wang et al., 2011*; *Yang et al., 2012*; *Kouser et al., 2013*; *Kloth et al., 2015*; *Bidinosti et al., 2016*; *Mei et al., 2016*; *Wang et al., 2016*). The various *Shank3*-deficient mouse models have displayed PMS and ASD-related behavioral phenotypes including impaired social behavior, increased repetitive behaviors, and motor deficits, as well as altered synaptic transmission and neuronal morphology in the brain (*Bozdagi et al., 2010*; *Peça et al., 2011*; *Wang et al., 2011*; *Yang et al., 2012*; *Kouser et al., 2013*; *Drapeau et al., 2014*; *Wang et al., 2016*). Recently, the translational relevance of these mouse models has been highlighted by our observation that the hormone IGF-1 improves motor and synaptic deficits observed in a *Shank3*-deficient mouse line (*Bozdagi et al., 2013*), a result which directly led to a safety and preliminary efficacy clinical trial of IGF-1 in children with PMS (*Kolevzon, 2014a*).

Here, we report the generation and characterization of the *Shank3*-deficient rat, representing a novel genetic model of PMS, which demonstrates clear PMS-related behavioral and electrophysiological phenotypes that can be ameliorated by intracerebroventricular (ICV) oxytocin administration.

# Results

## Production and developmental phenotyping of the Shank3-deficient rat model

Founder *Shank3*-deficient rats were generated using zinc-finger nucleases (ZFN) technology, targeting exon 6 of the ankyrin repeat domain. This domain was targeted because five patients had been described with mutations in it (*Figure 1A*) (*Durand et al., 2007*; *Moessner et al., 2007*; *Hamdan et al., 2011*; *Boccuto et al., 2013*; *Yuen et al., 2015*). Interestingly, the predicted truncated protein generated upon ZFN targeting of the rat *Shank3* gene is quite similar to one of the human mutations that have been described (*Hamdan et al., 2011*) (*Figure 1A*, middle sequence). In rat, this mutation leads to a significant reduction in the number of *Shank3* transcripts (*Figure 1—figure supplement 1A*) and reduces expression levels of the full-length Shank3a protein (*Figure 1B* and *Figure 1—figure supplement 1B*). We also observed that the levels of the PSD scaffolding protein Homer1 are decreased in the *Shank3*-deficient rats (*Figure 1—figure supplement 1C*), consistent with changes in the PSD and with well-replicated findings from *Shank3*-deficient mice. We noted that, at weaning, there was a modest reduction in the number of homozygous knockout (KO) animals from Heterozygous (Het) x Het matings, compared to expectation (292:555:200, corresponding to ratios of 0.53:1:0.36).

To evaluate the impact of *Shank3*-deficiency in rats, basic developmental processes as well as early motor and sensory function were assessed. These assessments were carried out as previously described (*Brunner et al., 2015*) and included weight, stomach milk content, body temperature, locomotion, grooming, rearing, pup ultrasonic vocalization, geotaxis, and the righting reflex. We found no genotype-related deficits in these basic developmental or functional processes. In addition, when tested on the elevated plus maze, *Shank3*-Het and KO rats did not exhibit increased anxiety-like behaviors (See Materials and methods for details and *Supplementary file 1* for results). These results enabled us to examine more complex PMS and ASD-relevant behaviors.

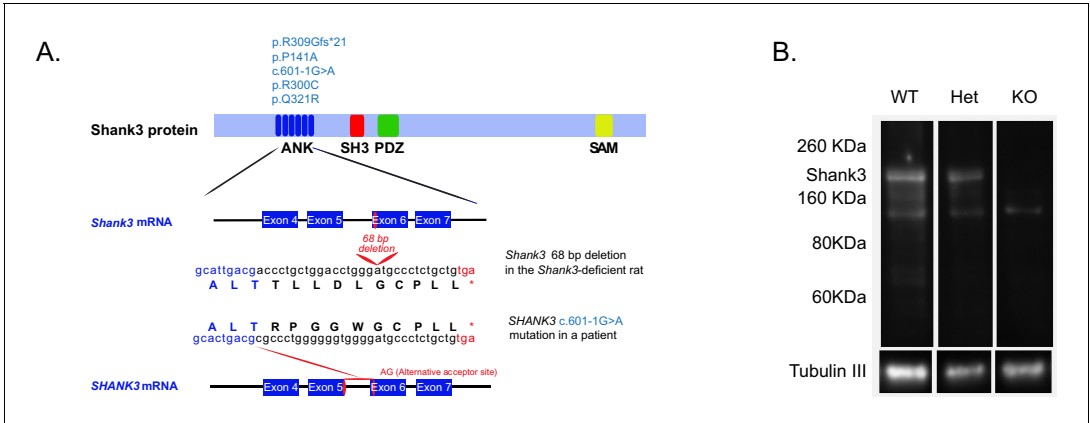

**Figure 1.** Gene-targeting of the rat *Shank3* gene. (**A**) The top schematic shows the Shank3 protein domains [ankyrin repeats domain (ANK), a Src homology 3 (SH3) domain, a PDZ domain, and a sterile α-motif (SAM) domain] and indicates the published de novo mutations observed within the ANK domain in PMS patients (light blue text, top schematic) (*Durand et al., 2007*; *Moessner et al., 2007*; *Hamdan et al., 2011*; *Boccuto et al., 2013*; *Yuen et al., 2015*). The 68 bp deletion that we introduced in exon 6 of the *Shank3* rat gene (middle schematic) produces a stop codon in exon six that truncates the Shank3 protein. In a PMS patient, the c.601–1G>A mutation in intron 5 of the *SHANK3* gene abolishes the normal acceptor site and leads to utilization of a cryptic acceptor site introducing a premature stop codon in exon 6 (red lines, bottom schematic), which also results in a similar truncated Shank3 protein (*Hamdan et al., 2011*). Lower-case letters, genomic sequence; upper-case letters, amino acids; *, premature stop codons. (**B**) Western blot showing anti-Shank3 staining, using antibodies targeted against the SH3 domain.

The following figure supplement is available for figure 1:

**Figure supplement 1.** The introduced mutation in rat targets exon 6 and leads to reduced overall *Shank3* transcripts and to decreased levels of the full-length Shank3 and Homer proteins.

## Social memory deficits in Shank3-deficient rats

We first measured preference for a social stimulus versus an object (*Figure 2—figure supplement 1A*), as well as juvenile social play (*Figure 2—figure supplement 1B*) and adult dyadic social interactions (*Supplementary file 1*) in freely interacting animals. We found no differences between *Shank3*-deficient and wildtype (WT) rats in any of these measures of social behavior or social preference.

Next, we used social habituation-dishabituation and social discrimination (SD) tests to examine social recognition memory (SRM). In the SD experiments, the subject rat was ultimately tested in its ability to discriminate between familiar and unfamiliar juvenile rats simultaneously introduced to it for 5 min. We used two versions of the SD test, in order to examine both short- and long-term SRM (with short and long referring to the interval between social memory acquisition and recall). To assess short-term SRM, the SD test was performed 30 min after the subject encountered the familiar rat for 5 min (*Figure 2A*), while, in the more challenging long-term SRM, which requires longer exposure to the social stimulus (*Gur et al., 2014*), the same test was performed 24 hr after a 1 hr encounter with the familiar rats (*Figure 2B*). We found that *Shank3*-deficiency does not impair short-term SRM as shown by the comparable findings between WT and *Shank3*-deficient rats on the social habituation-dishabituation (*Figure 2—figure supplement 1C*) and short-term SD tests (*Figure 2A*). In contrast, in two independent cohorts, we found that *Shank3*-Het and KO rats were unable to discriminate between novel and familiar social stimuli in the long-term SD test (*Figure 2B*) as they spent equal time investigating both the familiar and novel social stimuli. Notably, total investigation time (toward both familiar and unfamiliar rats) did not differ across genotypes in any of the SD tests (*Figure 2—figure supplement 1D*), indicating there is not a decreased interest in social exploration. Moreover, the fact that the *Shank3*-Het and KO rats performed well on the short-term SD test and were able to perceive and remember their conspecifics, even when only given 5 min for the first interaction, also ruled out perceptual deficits.

To determine whether this observed impairment is selective to social memory or if it also involves more general memory processes, we tested the rats on two long-term non-social memory paradigms that, similarly to the SD test, are known to be hippocampal-dependent, specifically, the object location memory test and the contextual fear conditioning memory test. In contrast to the impaired behavior in the long-term SD test, *Shank3*-Het and KO rats performed similarly to their WT littermates in the object location memory and contextual fear conditioning memory tests (*Figure 2C and D*). These results indicate that *Shank3* deficiency selectively impairs long-term social memory, but leaves intact both short-term social memory and non-social long-term memory.

## Attention deficits in Shank3-deficient rats

Attention deficits are often associated with PMS. Thus, we assessed performance in the attentionally demanding 5-choice serial reaction time (5-CSRT) task in which rats must respond quickly to briefly presented light cues (*Figure 3*) (*Mar et al., 2013*). This task requires training the rats in stages where the duration of the light stimulus is slowly decreased from 32 to 1 s by halving the stimulus duration across sessions once performance criteria are met (i.e. accuracy rates higher than 80% for two consecutive days with omission rates lower than 20%). *Shank3*-Het and KO rats learned the task and were able to reach baseline, similar to WT controls. However, both the *Shank3* Het and KO rats had lower accuracy and lower omission rates, when compared to WT rats, even after extensive training. Moreover, after reaching baseline criterion *Shank3*-deficient rats did not maintain even this level of performance across the 10-day test period, during which they performed significantly fewer correct trials (*Figure 3A*), made more errors (*Figure 3B*), and exhibited higher omission rates than WT rats (*Figure 3C*). Even on trials with a correct response, *Shank3*-deficient rats responded more slowly and with more variable latencies than WT rats (*Figure 3D*).

Slow, inaccurate and omitted responses to very brief visual stimuli are commonly interpreted as reflecting an attention deficit (*Robbins, 2002*). While changes in accuracy might also be attributed to deficits in sensory perception, we excluded this possibility by carrying out studies with less bright visual cues and *Shank3*-deficient rats performed at WT levels (not shown). It was only when the duration of the light cues was shortened that the deficits were manifested, which indicates impaired vigilance and/or spatial attention. Furthermore, these deficits were not due to decreased motivation for food, because the latency of *Shank3*-deficient rats to collect reward after a correct response was similar to WT rats (*Figure 3—figure supplement 1A*), as was task performance when light cues were

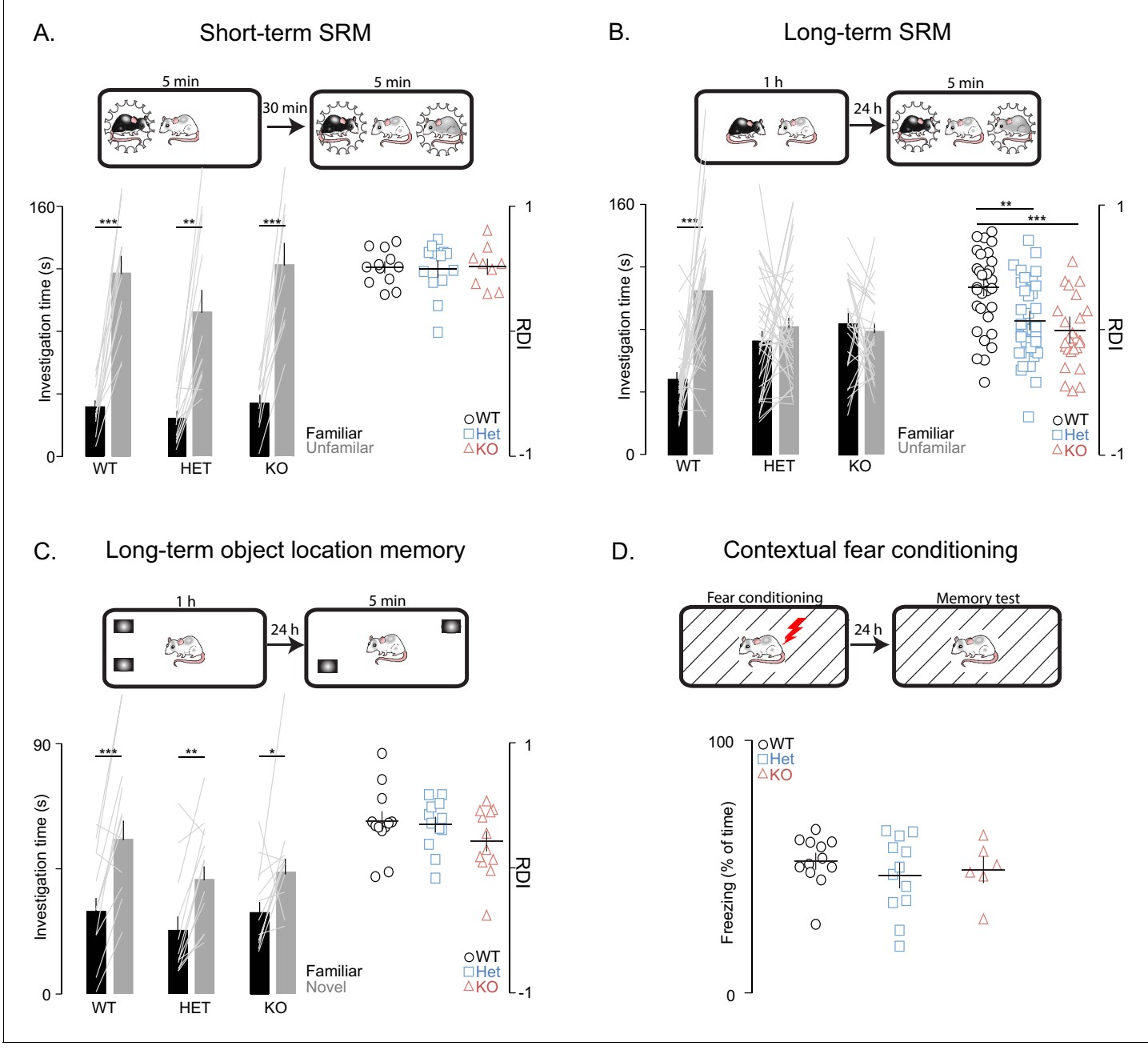

**Figure 2.** *Shank3*-deficient rats exhibit deficits in social memory. (**A–B**) Above each figure are schematics of the short- and long-term social discrimination (SD) paradigms. The examined adult subject is shown in white, and the juveniles are shown in black and grey. Bar plots show behavior in short-term (WT, n = 12; Het, n = 13; KO, n = 9) and long-term (WT, n = 31; Het, n = 37: KO, n = 25) SD tests. For long-term SD, similar results were observed in two independent cohorts, therefore the results were pooled and presented here as a single cohort. Bars ± SEM at left show test subjects average investigation time of a familiar and unfamiliar juvenile rat. The light overlaid gray lines in A and B show the corresponding individual subject data that comprise each bar. Right scatter plots, presented with mean ± SEM, show the ratio of the investigation time (RDI= (Unfamiliar-Familiar)/ (Unfamiliar+Familiar) for individual subjects. (**C**) Above the figure is a schematic of the long-term object location memory paradigm. The same plotting conventions as bar plots in A and B are used, but here they quantify investigation times of an object (WT, n = 12; Het, n = 12; KO, n = 12) in a novel or familiar location. (**D**) Above, Contextual fear conditioning paradigm schematic. Scatter plots (mean ± SEM) represent percent time freezing during retrieval of a 1-day-old conditioned fear memory (WT, n = 12; Het, n = 12; KO, n = 6). *, p<0.05, **, p<0.01, ***, p<0.001; see *Supplementary file 1* for detailed statistical results.

The following figure supplement is available for figure 2:

**Figure supplement 1.** General social behaviors are unaffected in *Shank3*-deficient rats.

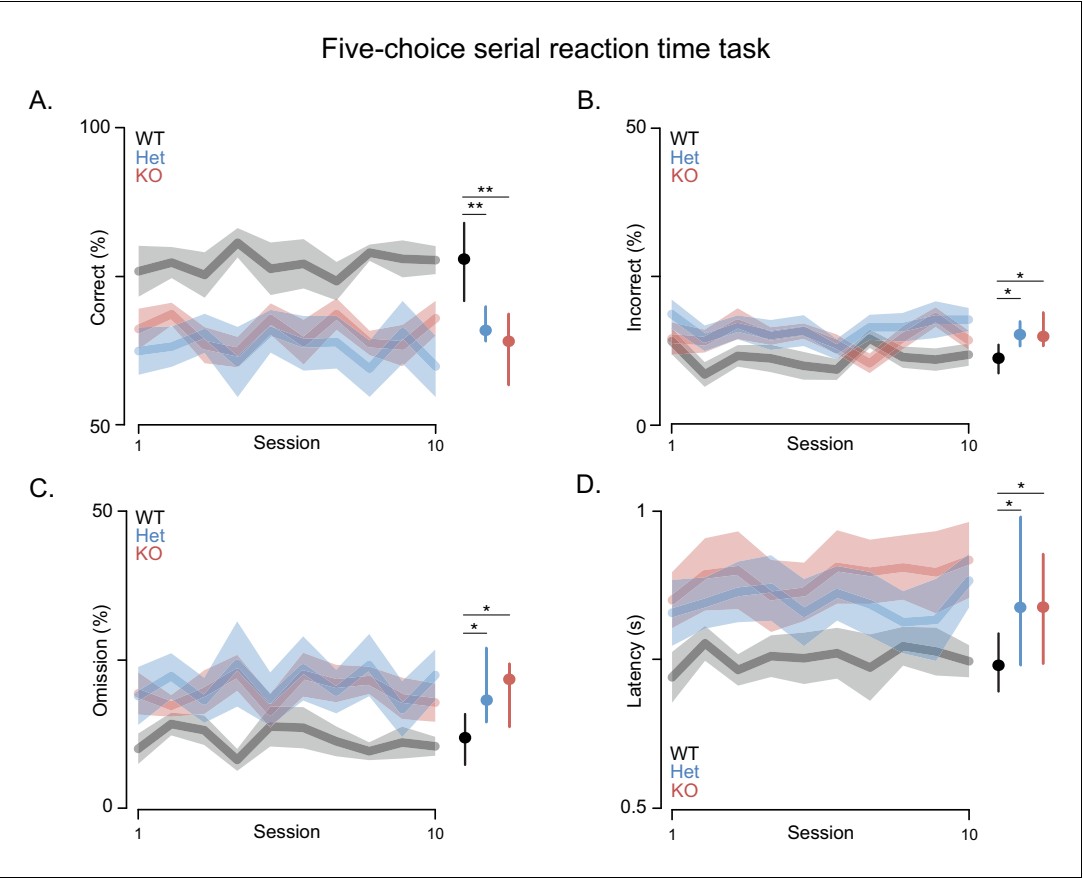

**Figure 3.** *Shank3*-deficient rats exhibit deficits in attention. (**A**) Traces and clouds indicate mean percentage of trials with a correct response ± SEM (WT, n = 10; Het, n = 13; KO, n = 12) across 10 5-CSRT sessions. The right side in all panels is the cross-rat median (dot) and middle quartiles (vertical lines). (**B**) Traces represent mean percentage of trials where an incorrect response was made. (**C**) Mean percentage of trials with no cued response. (**D**) Average reaction times on trials with a correct response. Results were observed in two independent cohorts; therefore, the results were pooled.

The following figure supplement is available for figure 3:

**Figure supplement 1.** *Shank3*-deficient rats exhibit normal motivation for reward and task performance is at WT levels when cue durations are long.

of a longer duration (*Figure 3—figure supplement 1B*). In summary, these results indicate that *Shank3*-deficient rats are impaired in an attentionally demanding task, which, when considered together with the deficits in long-term social memory, indicates this model demonstrates face validity for some features of PMS.

## Synaptic plasticity deficits in Shank3-deficient rats

Memory deficits, while not considered core symptoms of autism, have been associated with ASD (*Boucher et al., 2012*). Both impaired working (*Barendse et al., 2013*) and episodic memory (*Maister et al., 2013*) have been observed in human subjects with ASD, which has been attributed to aberrant connectivity of the hippocampus and medial prefrontal cortex (mPFC) (*Ben Shalom, 2003*). The behavioral deficits we observe in *Shank3*-deficient rats are consistent with dysfunction in these circuits. The hippocampus and mPFC are both important for SRM (*Watson et al., 2012*; *Harvey and Lepage, 2014*; *Jacobs and Tsien, 2014*). Moreover, performance in the 5-CSRT task depends on mPFC function (*Rogers et al., 2001*) and attention, working memory, and decision-making require intact hippocampal-prefrontal functional connectivity (*Jones and Wilson, 2005*). We therefore evaluated the effect of *Shank3* deficiency on synaptic function and plasticity in hippocampal-PFC circuitry. Extracellular field excitatory postsynaptic potential (fESPs) recordings at Schaffer collateral-CA1 synapses were similar between genotypes, with no differences in paired-pulse

facilitation (*Supplementary file 1*) or the input-output relationship (*Figure 4—figure supplement 1*). These results suggest that basal synaptic transmission is generally intact in *Shank3*-deficient rats.

We found, however, in independent cohorts, that plasticity at these synapses was not intact. Long-term potentiation (LTP) induced by high-frequency stimulation (HFS) was reduced in both *Shank3*-Het and KO rats (*Figures 4A and 6A*, and *Figure 6—figure supplement 1A*), while mGluR-dependent long-term depression (LTD) was reduced only in KO rats (*Figure 4B*).

Innervation of the mPFC by the hippocampus (*Marquis et al., 2006*) is important for attention and working memory, both of which we have shown here are impaired in *Shank3*-deficient rats. We therefore assessed hippocampal-PFC synaptic transmission in vivo by stimulating hippocampal CA1/subicular regions and recording fEPSPs in the prelimbic area of the PFC in anesthetized rats. We found that LTP was reduced in both *Shank3*-Het and KO rats (*Figure 4C*). We also observed that there were no genotype associated differences in the input-output relationship of evoked local field potentials in coronal PFC slices, which demonstrates that these changes were specific to hippocampal-prefrontal circuitry (*Supplementary file 1*). In summary, these results suggest that *Shank3*-deficiency impairs plasticity in both the projections from hippocampus to the PFC and within intrinsic hippocampal circuits.

## Oxytocin improves behavior and synaptic plasticity deficits in Shank3-deficient rats

Oxytocin has a central role in SRM formation (*Ferguson et al., 2000*; *Gur et al., 2014*) and hence we reasoned that it may underly some of the altered behaviors we observed; in addition and more broadly, oxytocin modulates mammalian social behavior and may be dysregulated in ASD (*Harony and Wagner, 2010*). To determine whether the behavioral deficits we observed in *Shank3*-deficient rats could be improved with a pharmacological intervention strategy, we tested *Shank3*-deficient rats on long-term SD and 5-CSRT tasks following an injection of either oxytocin or saline into the left lateral ventricle. We observed that oxytocin improved both the long-term social memory and attention deficits in *Shank3*-Het and KO rats (*Figure 5A and B*). Notably, in WT animals,

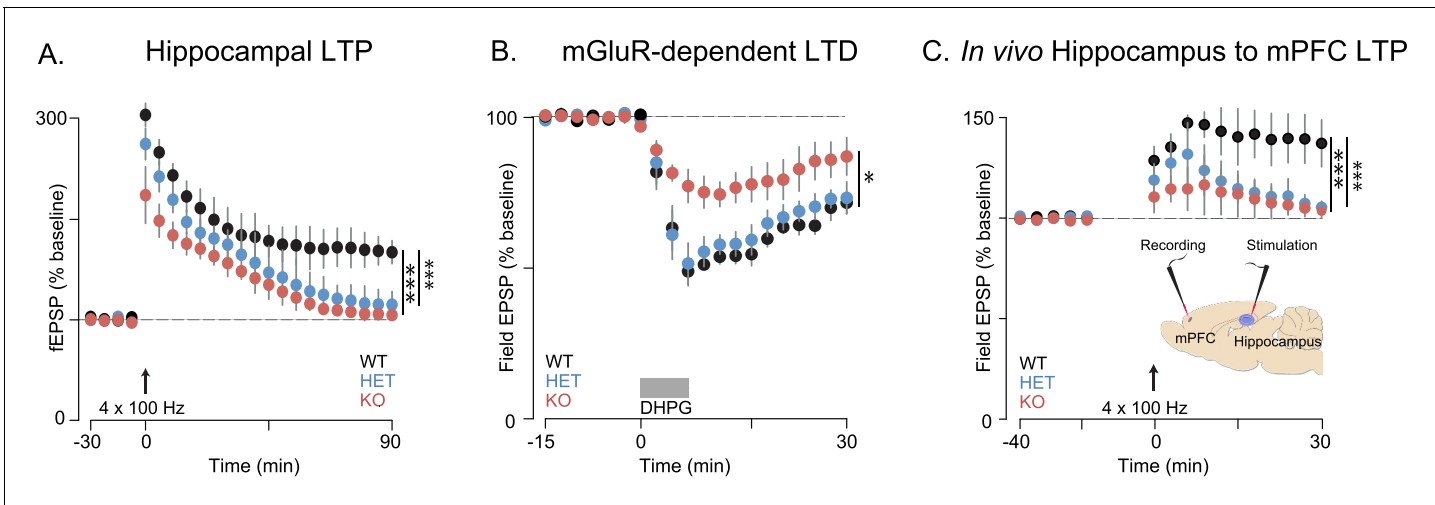

**Figure 4.** Synaptic plasticity is impaired in *Shank3*-deficient rats. (**A**) High-frequency stimulation (HFS, arrow)-induced long-term potentiation (LTP) at hippocampal Schaffer collateral-CA1 synapses (n = 6 rats/genotype, 1–2 slices per rat). (**B**) Long-term depression induced by the mGluR agonist DHPG (50 μM, 5 min) is indicated by the horizontal line (n = 6 rats/genotype, six slices per rat). (**C**) HFS-induced LTP in the prelimbic PFC after stimulation of ipsilateral CA1 in ventral hippocampus of intact anesthetized WT (n = 5), *Shank3* Het (n = 6) and KO (n = 6) rats. Inset shows a schematic of the target location of the stimulating and recording electrodes in vivo. Summary data are presented as mean ± SD. *p<0.05, ***p<0.001; See *Supplementary file 1* for detailed statistical results.

The following figure supplement is available for figure 4:

**Figure supplement 1.** Basal synaptic transmission is intact in *Shank3*-deficient rats.

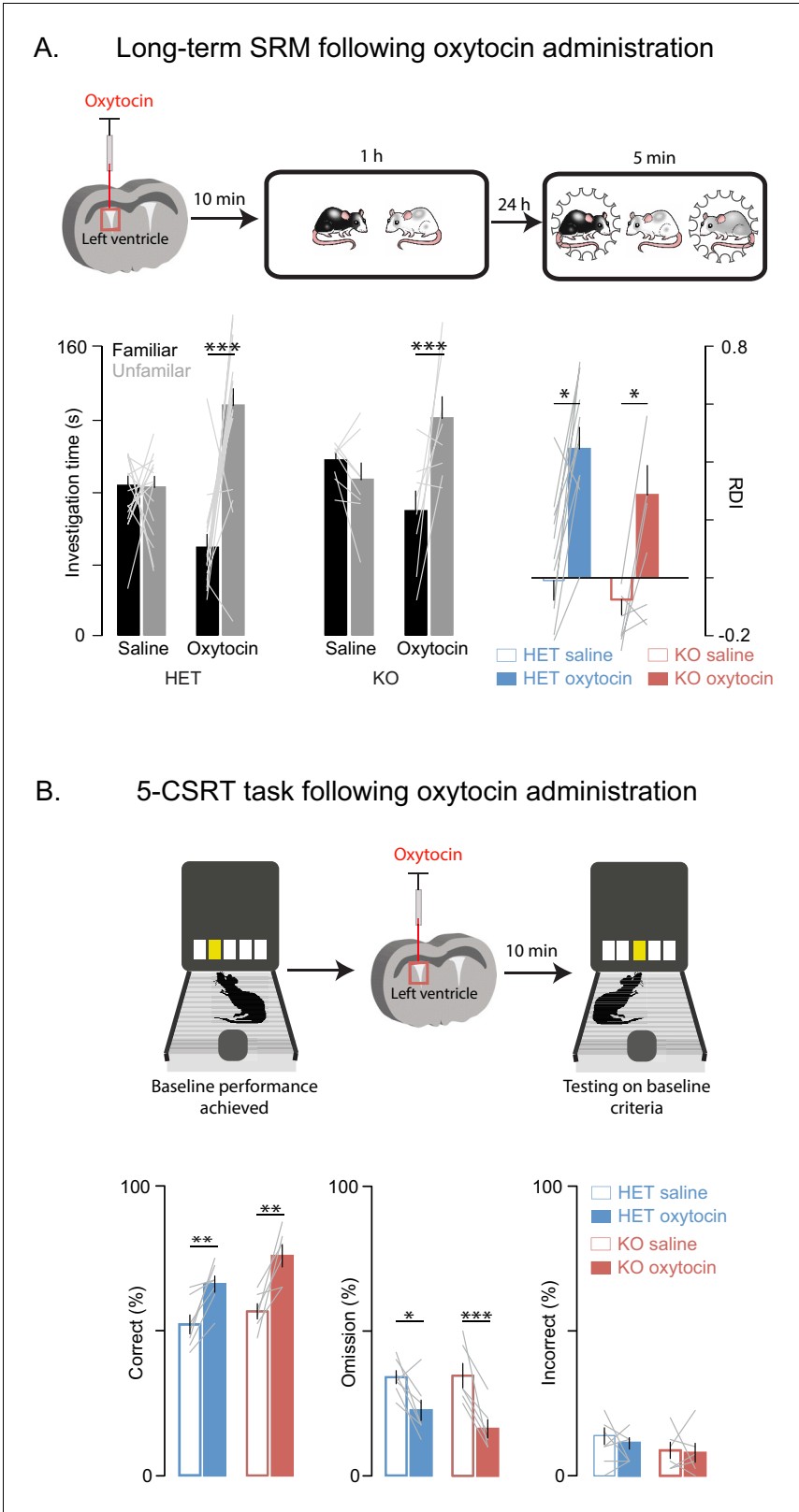

**Figure 5.** Oxytocin improves social memory and attentional deficits in *Shank3*-deficient rats. Above each figure is a schematic depicting the sequence of oxytocin administration and behavioral testing on the long-term social discrimination (SD) paradigm (in **A**) or 5-CSRT task (in **B**). In **A**, the test subject is shown in white and the juveniles are shown in black and grey. (**A**) Left bars (± SEM) show the test subject's average investigation time of a familiar

*Figure 5 continued on next page*

*Figure 5 continued*

and unfamiliar juvenile rat on long-term SD test following treatment with oxytocin or vehicle for *Shank3* Het (n = 15) and KO (n = 8) rats. The light overlaid gray lines show the corresponding individual subject data that comprise each bar. Bars on right show the ratio of the investigation time (RDI= (Unfamiliar-Familiar)/(Unfamiliar +Familiar) for individual subjects with the light overlaid gray lines representing the corresponding individual subject data. (**B**) Bars (± SEM) represent the percentage of correct (left), omitted (middle), and incorrect (right) trials in saline (solid bars) and oxytocin (open bars) of *Shank3* Het (n = 7) and KO (n = 6) rats. Color conventions are identical to those used in *Figure 3A–C*. The light overlaid gray lines show the corresponding individual subject data that went into each bar. *p<0.05, **p<0.01, ***p<0.001; See *Supplementary file 1* for detailed statistical results.

The following figure supplement is available for figure 5:

**Figure supplement 1.** Oxytocin has no effect on social memory or attention in WT rats.

---

oxytocin had no effect on long-term SRM (*Figure 5—figure supplement 1A*) or 5-CSRT task performance (*Figure 5—figure supplement 1B*).

Since oxytocin reversed the behavioral deficits in *Shank3*-deficient rats and has been previously shown to enhance the induction of synaptic plasticity in several systems (*Benelli et al., 1995*; *Tomizawa et al., 2003*; *Fang et al., 2008*; *Ninan, 2011*; *Lin et al., 2012*; *Gur et al., 2014*), we also examined the effect of oxytocin on LTP both in vitro and in vivo. As previously reported, in acute hippocampal slices derived from WT rats, oxytocin enhanced LTP induction after a weak stimulation of one train of 100 Hz pulses (*Tomizawa et al., 2003*; *Lin et al., 2012*) but we did not observe this oxytocin-dependent enhancement in *Shank3*-deficient rats (*Figure 6—figure supplement 1A*). In WT slices, oxytocin had no effect on LTP induced by stronger stimulation (4 × 100 Hz stimulation trains), but it greatly enhanced LTP in slices prepared from *Shank3*-deficient rats (in independent cohorts, *Figure 6A* and *Figure 6—figure supplement 1B*). In vivo, oxytocin also reversed the impaired LTP at hippocampal-prefrontal synapses in *Shank3*-deficient rats (*Figure 6B*). These results indicate that oxytocin treatment restores LTP induction at hippocampal and hippocampal-prefrontal

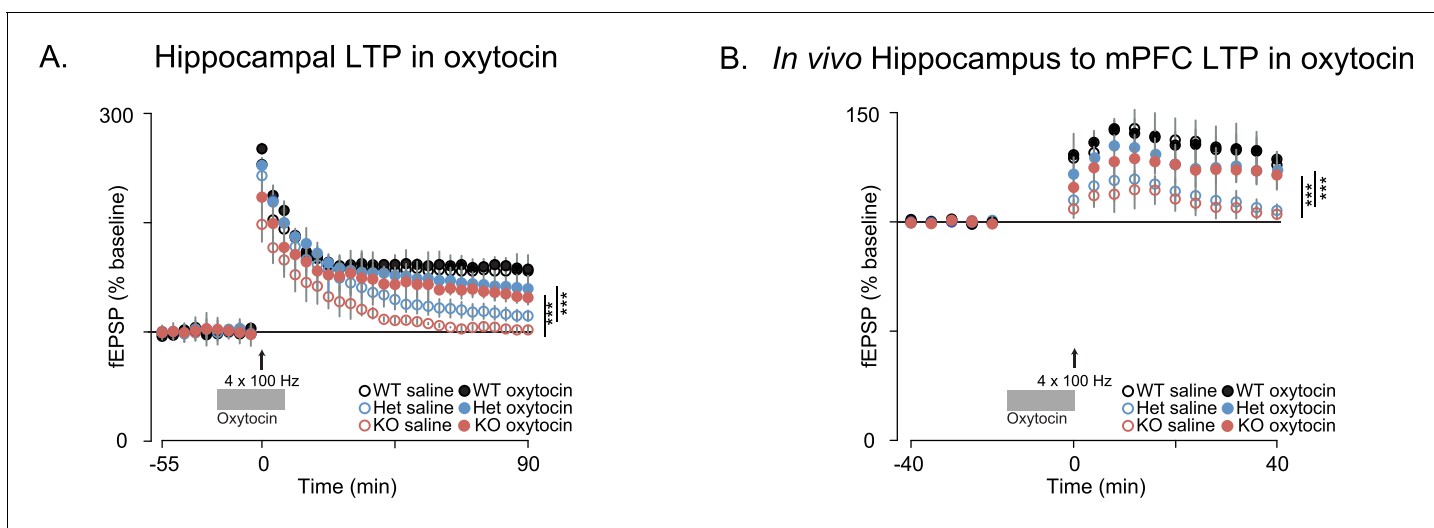

**Figure 6.** Oxytocin improves synaptic plasticity deficits in *Shank3*-deficient rats. (**A**) Traces depict Hippocampal HFS-induced LTP (4 × 100 Hz) in WT and *Shank3*-deficient rats (n = 4 rats/genotype, six slices per rat). Application of 1 μM of oxytocin is indicated by the horizontal line. (**B**) LTP in the hippocampal to prefrontal synaptic pathway recorded in vivo in WT (n = 3), *Shank3* Het (n = 4) and KO (n = 4) rats. Oxytocin (2 ng) or saline was administered in the lateral ventricle 5 min before LTP induction. ***p<0.001. See *Supplementary file 1* for detailed statistical results.

The following figure supplement is available for figure 6:

**Figure supplement 1.** Oxytocin enhances LTP induced by a single pulse at 100 Hz in WT but not in *Shank3*-deficient rats.

synapses in *Shank3*-deficient rats and is consistent with a mechanism whereby deficits in synaptic plasticity across hippocampal-prefrontal circuitry underly PMS-relevant behavioral deficits we observed in these animals.

## Discussion

Deficits in attention and the presence of ASD-related behaviors are common in PMS. In our genetically modified rat model of PMS, we found that *Shank3* deficiency impairs social memory and attention. These impairments are accompanied by attenuated synaptic plasticity within the hippocampus and in the monosynaptic pathway connecting the hippocampus to the PFC. Hippocampal synaptic plasticity deficits have also been reported in *Shank3*-mouse models and were associated with impairments in actin cytoskeleton remodeling and with changes in the levels of glutamatergic receptors and other PSD scaffolding components including Homer, which we also found to be decreased in the *Shank3*-deficient rat (*Figure 1—figure supplement 1C*) (*Bozdagi et al., 2010*; *Wang et al., 2011*; *Duffney et al., 2013*; *Kouser et al., 2013*; *Wang et al., 2016*). The fact that long-term social memory was selectively impaired, in contrast to more general memory processes, may reflect the complexity of the information involved in social memory (*Adolphs, 2010*; *Wiley, 2013*), which may require more elaborate synaptic mechanisms than simpler forms of memory. These results may also relate to previous findings that correlate ASD-associated episodic memory deficits with the complexity of the memoranda (*Lind, 2010*), which suggests that sub-optimal thresholds for synaptic plasticity interfere with the ability of complex stimuli, such as social stimuli, to activate memory formation. Deficits in attentionally demanding tasks, like the 5-CSRT task, may also result from blunted synaptic plasticity, where brief events must be rapidly encoded and reliably stored by neural circuits to promote appropriate and timely behavior.

It is interesting, and important, to compare findings in different rodent models of PMS and relate them to symptoms observed in patients. In particular, while there is some overlap between the social communicative deficits observed in PMS and those observed in what is considered more classical autism, there are important differences. In contrast to the more typical conception of ASD (whether accurate or not), PMS-associated ASD is *not* associated with a social aversion or lack of social approach (*Soorya et al., 2013*; *Kolevzon, 2014b*). A recent study of functional neuroimaging showed that differential fMRI changes in response to social versus non-social sounds are preserved in PMS, in contrast to studies of idiopathic autism (iASD) (*Wang et al., 2016*). These findings, together with the fact that not all individuals with PMS are diagnosed with ASD, indicate that animal models for PMS should not necessarily present with social behavioral deficits and imply that behavioral phenotypic diversity in PMS patients may reflect additional processes.

This may explain some of the discrepancies that have been reported across distinct *Shank3*-mouse models (*Harony-Nicolas et al., 2015*), and those that we report here in the *Shank3*-deficient rat. These discrepancies could be attributed to (1) mutations/deletions in existing rodent models were targeted to different parts of the *Shank3* gene, (2) varying genetic background, (3) different behavioral paradigms, (4) varying age and sex, (5) a focus on heterozygotes or knockouts, or other factors. Beyond these methodological issues, from an evolutionary perspective, mice and rats are separated by millions of years, which likely explain many of the observed discrepancies in behavioral repertoires between the two species. Just as mice and rats differ, so do mice and rats greatly differ from human, and thus one should be careful not to overly anthropomorphize these model systems or look for perfect overlap across species.

Further investigation of the mechanisms underlying synaptic plasticity deficits and the molecular pathways affected by *Shank3* mutation and the consequent synaptic changes in rat and mouse *Shank3* models will likely provide new targets for therapeutic treatments and allow comparison between two different species, thus providing more reliable molecular targets for future drug development studies. Although all the mutations that have been studied, including the one we introduced to the rat model, lead to the deletion of the longest *Shank3* isoform, most leave intact other shorter isoforms, whose role is still unclear. Because many patients lack the entire *SHANK3* gene, understanding the role of different Shank3 isoforms is of great interest, and contrasting existing rodent models will be useful in clarifying this important question.

To date, treatment strategies for PMS are non-specific and do not address key issues of cognition, language, motor development, etc. There are approaches that are applied in PMS that are

adopted from the general population. For example, when ASD is present, one might make use of behavioral interventions (*Kolevzon et al., 2014b*), but these strategies need to be improved and designed for the specific needs of PMS patients. Recently, the field has begun to translate basic neurobiological findings gleaned from rodent models into promising pharmacological treatments for a host of genetic disorders such as Fragile X syndrome, Tuberous sclerosis, Rett syndrome, and PMS (*Bear et al., 2004*; *Ehninger and Silva, 2009*; *Tropea et al., 2009*; *Kolevzon et al., 2014a*).

The peptide oxytocin is a powerful regulator of mammalian social behavior, which has been shown to improve various aspects of social cognition and social behavior in human and non-human primates, by increasing social memory, enhancing social reward, and modulating social attention (*Chang et al., 2012*; *Guastella et al., 2012*; *Parr et al., 2016*). Oxytocin has been also studied in clinical trials as a potential therapeutic for iASD-associated social impairments, yet these efforts have led to equivocal results (*Guastella et al., 2016*). Hitherto, the effect of oxytocin has not been investigated in PMS or in animal models of PMS, and its effect on non-social ASD and PMS-associated deficits has not been evaluated. Given the evidence for a key role for oxytocin in SRM, we tested this compound and observed important effects on both behavioral and electrophysiological deficits. Interestingly, an ameliorative effect of oxytocin treatment on autism-like behaviors has recently been reported in other genetically defined mouse models of ASD (*Tyzio et al., 2014*; *Peñagarikano et al., 2015*).

Our results show, for the first time to our knowledge, a beneficial effect of oxytocin on attention to non-social stimuli. Social and attention deficits are often co-morbid in PMS (*Soorya et al., 2013*; *Kolevzon, 2014b*), which is perhaps reflective of a shared etiology and may explain why oxytocin improved both deficit types in *Shank*3 deficient rats. The ameliorative effect of oxytocin treatment on the long-term social memory, attention and synaptic plasticity deficits in *Shank3*-deficient rats, support a mechanism whereby deficits in synaptic plasticity across hippocampal-prefrontal circuitry may underly the behavioral deficits we observed. These results also imply that exogenous oxytocin administration might have therapeutic potential in human PMS patients.

In our view, genetically modified rat models are especially valuable for behavioral and functional studies. They have several species-specific advantages over mouse models, including a more complex behavioral repertoire and larger brains that readily facilitate high-density electrophysiological recordings. Moreover, rats remain the primary choice of the pharmaceutical industry for studying the pharmacokinetic properties of novel drugs that may have therapeutic potential in human. Our production of a *Shank3*-deficient rat model, which demonstrates both construct and face validity, will pave the way for future studies that investigate the mode by which *Shank3* mutations alter brain activity during behavior and to further study the effects of oxytocin and other potential therapeutics for PMS. Our findings open new avenues of research to study the effect of *Shank3*-deficiency on the development and function of oxytocin neurons. Future studies should investigate if *Shank3*-deficiency affects hypothalamic processes that regulate oxytocin production and secretion (*Brownstein et al., 1980*) or whether it perturbs pathways further downstream. Additionally, it would be important to determine if oxytocin treatments during critical developmental windows could ameliorate PMS-associated behavioral deficits, and whether other forms of ASD, defined either etiologically or by the presence of specific biomarkers, may show such broad response to oxytocin. These, and other experiments, would support the use of oxytocinergic agonists for multiple forms of ASD, including additional genetically defined forms of ASD.

## Materials and methods

### Generation of Shank3-deficient rats

*Shank3*-deficient rats were generated using zinc-finger nucleases (ZFN) on the outbred Sprague-Dawley background. The design, cloning, and validation of the ZFN, as well as embryonic microinjection and screening for positive founder rats was performed by SAGE Labs (Boyertown, PA). Briefly, sixteen pairs of ZFN were designed targeting exons 4, 5 and 6 of the *Shank3* gene to disrupt the *Shank3* ANK domain. These pairs were assembled by PCR and sub cloned into the pZFN expression plasmid. ZFN were then transfected into the rat C6 cell line and tested for disruption activity using the Surveyor endonuclease (CEL-1) assay (*Kulinski et al., 2000*). The best performing ZFN pair GACGCCCCTGTACCATAGTgccctaGGGGGCGGGGATGCC (recognizing the GACGCCCCTG

TACCATAGT and the GGGGGCGGGGATGCC sequences located between 130163213–130163231 and 130163238–130163252, respectively, in the *Shank3* gene (NCBI reference sequence NC_005106.3), were used for embryo microinjection. Positive Sprague-Dawley founder animals were mated to produce F1 heterozygous breeder pairs. Genomic DNA sequencing using the ZFN Primer F (GGAGGGACTCAATGCAGAAA) identified a 68 bp deletion in exon six leading to a premature stop codon, as shown in *Figure 1A*.

### Animal care and husbandry

All rats were kept under veterinary supervision in a 12 hr light/dark cycle at 22 ± 2°C. Unless otherwise indicated, animals were pair-caged with food and water available *ad libitum*. All animal procedures were approved by the Institutional Animal Care and Use Committees at the Icahn School of Medicine at Mount Sinai, the University of Haifa, and the Albert Einstein College of Medicine.

### Animal selection and inclusion criteria

Age-matched males were used in all tests unless otherwise noted. Minimal group sample sizes were decided in advance of the described experiments based on the standards in the field and using on-line sample size/power calculation tools, specifically the 'biomath' online tool; (http://biomath.info/power/). Any animals excluded from analysis or testing due to poor performance or for any other reason are noted in the appropriate Methods section.

### Molecular analysis

#### Isolation of postsynaptic density (PSD)

PFC tissues were dissected from 8 week old rats as previously described (*Spijker, 2011*). PSDs were prepared as follows; tissue samples from WT and Shank3 KO rats were homogenized in 2-[4-(2-hydroxyethyl)piperazin-1-yl]-ethanesulfonic acid (HEPES)-A containing 4 mM HEPES, pH 7.4, 0.32 M sucrose, Protease Inhibitor Cocktail and PhoSTOP Phosphatase Inhibitor Cocktail (both from Roche). Nuclear fractions were then precipitated by centrifuging twice at 700 *g* for 15 min, and the resulting supernatants were further centrifuged at 21,000 *g* for 15 min. The precipitates were re-suspended in HEPES-B containing 4 mM HEPES, pH 7.4, Protease Inhibitor Cocktail and PhoSTOP Phosphatase Inhibitor Cocktail, homogenized and rotated at 4°C for 1 hr. These lysates were centrifuged at 32,000 *g* for 20 min and washed twice with HEPES-C containing 50 mM HEPES, pH 7.4, 0.5% Triton X-100, Protease Inhibitor Cocktail and PhoSTOP Phosphatase Inhibitor Cocktail. Finally, postsynaptic density fractions were resuspended in HEPES-C containing 1.8% sodium dodecyl sulfate (SDS) and 2.5 M urea. Protein concentrations were determined using the Pierce BCA protein assay.

#### Immunoblotting

The immunoblotting experiments, presented in *Figure 1B* and *Figure 1—figure supplement 1B and C*, were performed using a standard protocol (*Bozdagi et al., 2010*). Briefly, 10 µg of each PSD fraction were loaded onto a 4–12% SDS-polyacrylamide gel electrophoresis (PAGE gel, Invitrogen; Carlsbad, CA), which was then transferred to a polyvinylidene fluoride membrane for immunoblotting. For Shank3 protein, beta III tubulin, and Homer detection, anti-Shank3 (1:100, NeuroMab; clone N69/46, BRID:AB_10698031 targeted against amino acids 840–857), anti-Shank3 (1:100, NeuroMab; clone N367/62, BRID:AB_2315920 targeted against the SH3 domain), anti beta III tubulin (1:2000, Abcam; ab18207, RRID:AB_444319), and Homer-1b/c (1:100, Santa Cruz; sc-25271, RRID: AB_675652) antibodies were used. HRP-conjugated anti-rabbit (1:5,000), BRID:AB_2337910 HRP-conjugated anti-mouse antibodies (1:5,000), BRID:AB_2340031 were purchased from Jackson ImmunoResearch Laboratories (West Grove, PA). ECL substrate (Pierce; Thermo Scientific, Rockford, IL) or SuperSignal West Femto (Thermo Scientific, Rockford, IL) substrates were used to produce the signal that was detected on a G:Box Chemi-XT4 GENE*Sys* imager (Syngene; Cambridge, UK). Blots were quantified using the software GeneTools (SynGene, version 4.02).

#### RNA sequencing and analysis

RNA was extracted from mPFC tissues from 8 week old WT and *Shank3*-KO littermate rats using the RNeasy Mini Kit (Qiagen, CA) according to the manufacturer's instructions. One µg was then used for the preparation of the seq library using TruSeq mRNA Seq Kit supplied by Illumina (Cat # RS-

122–2001), following the manufacturer's instruction. Before starting the seq library preparation, a Poly-A-based mRNA enrichment step was carried out and cDNA was synthesized and used for library preparation using the Illumina TruSeqTM RNA sample preparation kit as previously described (*Tariq et al., 2011*), except for the following steps: adapter-ligated DNA fragments were size-selected by gel-free size selection using appropriate concentration of SPRI AMPure beads to get an average 200 bp peak size in adaptor-ligated DNA. The size selected adaptor-ligated DNA fragments were amplified by LM-PCR. Then, Illumina recommended 6 bp barcode bases were introduced at one end of the adaptors during PCR amplification step. The amplified PCR products were then purified with SPRI AMPure XP magnetic beads to get the final RNA-seq library, which was used for high-throughput RNA-seq. Size and concentration of the RNA-seq library were measured by Bioanalyzer and Qubit.

We used standard approaches with 40 million 100 nt paired end sequences to reliably assess expression for each sample using the Genome Analyzer IIx (Illumina). The design of the experiment was as follows: 12 barcoded samples, 6 of each genotype, were pooled and loaded on two lanes, so that each sample was spread over two lanes to further minimize confounds, specifically those associated with lane effects. RNAseq analysis was carried out as previously described (*Tariq et al., 2011*), although with some modifications. Briefly, following the removal of sequences that mapped to ribosomal RNA sequencing, the remaining reads were mapped to the rat reference genome rn4, using TopHat 2.0.9 (*Trapnell et al., 2009*) and Bowtie 2.1.0 (*Langmead et al., 2009*), using default values. The mean insert sizes, as determined by bioanalyzer, were employed in TopHat mapping. Only uniquely mapped reads were considered and used in downstream analysis. Transcript abundance was determined using the HTSeq tool. For quality control, we used the Fastqc and the RNA-seqQC tools, to assess the quality of the raw data and the mapped reads, respectively.

To visualize the deletion in exon 6 of the *Shank3* gene, we used the Integrated Genome Browser (IGB), available online (http://bioviz.org/igb/). The *Rattus norvegicus* reference genome version Nov_2004 was selected, and the BAM alignment file, from each rat, was loaded separately (*Figure 1—figure supplement 1A*).

## Developmental studies

Pregnant *Shank3*-Het rats were shipped to PsychoGenics (Tarrytown, NY) where they were checked daily for litters. On the day of birth (defined as postnatal day 0 [P0]), the dam and her litter were left undisturbed. On P2, pups were tattooed using non-toxic ink applied under the skin of their paw and a tail snip sample was taken for genotyping. All pups were weighed daily from P2 to P21. Some animals were assessed for milk content score from P2 until their abdomen was covered by fur (around P10) to look for genotypic differences in the ability to breast-feed. The milk score for each pup within a litter is dependent on the overall level of milk in all pups from that litter at the time of observation: 3, normal milk content, normal milk level is defined in relation to littermates' milk content; 2, stomach not full, but milk is easily detected; 1, trace amounts of milk; 0, absence of milk. Additionally, the age at which the animals' eyes open was documented. Animals were checked for survival once per day and survival rates were recorded. Neonatal phenotyping assessments, including body temperature, isolation tests (measuring frequencies of square crossing, pivot, grooming, rearing, ultrasonic vocalization, sniffing, geotaxis, and righting reflex were conducted at either P7 or P15 as previously described (*Brunner et al., 2015*). All tests took place between 10:00 AM and 3:00 PM and were performed by an examiner blind to subject genotype (*Supplementary file 1*). At P15, animals were tested in the geotaxis and righting reflex tests.

### Negative geotaxis

Animals were placed on an inclined platform (approximately 35° incline) with their head facing down slope. The latency to reorient the head toward the higher end of the platform was recorded (*Supplementary file 1*). This test measures motor coordination and vestibular system function. Animals had a maximum of 60 s to complete the test.

### Pre-righting reflex and righting reflex

These motor coordination assays measure the ability of a rat to right itself (reorient itself back onto its feet) when placed on its back. The pre-righting reflex assessment scores the time (from 0 to 30 s)

it takes for the animal to right itself. Immediately after the pre-righting reflex test, 10 consecutive righting reflex trials (30 s each) were conducted and the number of successful righting trials (out of ten 30 s trials) was recorded (see *Supplementary file 1*).

## Behavioral tests

### Elevated plus maze

Anxiety levels of the rats were assessed at 6–8 weeks of age using the elevated plus maze (EPM) apparatus by PsychoGenics. The EPM apparatus (Kinder Scientific, Poway, CA) consists of four arms of equal dimensions at a 90° angle, connected via a 6 × 6 cm square area. Two of the arms (50 × 10.8 cm) have high walls (15.75 cm) on all sides except the central entry area ('closed arms'), and the other two arms have no walls ('open arms'). All surfaces visible to the experimental subjects are made of black acrylic. The whole apparatus is elevated approximately 85 cm above the floor and is surrounded by black vinyl curtains. Appropriate padding was placed underneath the EPM apparatus in the rare event that an animal fell when in the open arms of the maze. On test days, rats were habituated to the experimental room at least 1 hr prior to being exposed to the EPM. After this habituation period, the animals were placed in the center area of the EPM apparatus facing an open arm and allowed to freely explore for 5 min. We measured total distance traveled, time spent in and number of entries into both the open and closed arms of the EPM (see *Supplementary file 1* for a summary of these results). Testing and measurements were performed by an examiner blind to subject genotype.

### Adult dyadic social interaction

At 6–8 weeks of age, animals were tested for social behaviors by PsychoGenics. The adult dyadic social interaction test chamber is made of white Plexiglas (60.96 cm x 43.8 cm x 20.32 cm). Prior to measuring social behavior, all rats were habituated to the test room for at least 30 min. After this habituation period, test rats were placed in the chamber with strain-, sex-, and age-matched controls. Social interactions were recorded for the duration of the 6-min test session. Interactions were scored as social if they included sniffing, climbing over or under, following or exploring the anogenital area of the conspecific rat. Passive or aggressive contacts were not considered social interactions. Total social interaction time was measured and each pair of rats was treated as a single unit in order to avoid false inflation of group size (*Supplementary file 1*). Interactions were scored by an experienced observer who was blind to genotype.

### Social preference

We used a modified version of the widely used social preference paradigm by Moy et al., (*Moy et al., 2004*). Briefly, following 1 hr of habituation in the experimental cage, 9–11 week old rats were exposed simultaneously to a non-social object (e.g., hole puncher, paperclip, Rubik's cube) and to a juvenile rat confined to a transparent plastic corral. The total amount of time that the subjects spent investigating the social and non-social stimuli was measured by an experienced observer who was blind to genotype and the results are show in *Figure 2—figure supplement 1A*.

### Juvenile social play

Social play behavior, in WT and *Shank3*-KO rats, was measured at 26–30 days of age, in a manner similar to what has been previously described (*Hamilton et al., 2014*). Briefly, after weaning at 21 days of age, rats were pair-housed. Prior to the juvenile social play test each rat was individually habituated to the test cage for 10 min for three consecutive days. After this third habituation session, rats were socially isolated for 24 hr before the juvenile social play test began. On test days, two non-littermate rats, of the same gender, were placed into the 45×45 × 40 cm test arena for 10 min. The play partner was always a WT rat with no more than 10 g difference in body weight when compared to the test subject. The behavior of the animals was videotaped and analyzed post hoc using video tracking software (Ethovision XT, Noldus). Several aspects of social play were assessed, including approaching, chasing, crawling, nape attacking, pinning, and boxing of the conspecific rat. Other forms of social, but non-play behaviors, were also measured such as licking, grooming, and body or anogenital sniffing of the play partner. Approaching was included in both play and non-play social behaviors as it cannot be considered solely as part of the play sequelae. Behaviors were scored

independently by two trained observers who were blind to the test animal's genotype (*Figure 2—figure supplement 1B* and *Supplementary file 1*).

## Habituation-dishabituation social recognition memory

The habituation-dishabituation social recognition memory test was also carried out as previously described (*Spiteri and Agmo, 2009*), but with several modifications. Briefly, following 1 hr habituation to the experimental cage (60 × 40 × 20 cm), 9–11 week old male test rats were exposed for 5 min to the same social stimulus over four consecutive sessions separated by intervals of 30 min. In a fifth session they were exposed to a novel social stimulus of a different strain. In each session, the duration of investigation was measured, including any contact between the subject's nose and the juvenile's body or any other investigatory behavior displayed by the adult subject was measured by a trained observer, who was blind to genotype. Juveniles (3 week old , 30–35 g) male Wistar-Hanover or Wistar-Hola rats were used as social stimuli (*Figure 2—figure supplement 1C*).

## Long- and short-term social discrimination tests

Long- and short-term social discrimination tests were performed as previously described (*Gur et al., 2014*). Briefly, in the long-term SRM, the test rat (9–11 week old) was unrestrictedly exposed to a novel juvenile (3 week old, 30–35 g) of a different strain (Wistar Hannover/Hola) for 1 hr in an experimental cage (60 × 40 × 20 cm). The test rat was placed alone, 24 hr later, in the same test cage, but which now contained two transparent plastic corrals (9 cm in diameter). These corrals contain five horizontal slots per side (1 × 13 cm each), which allow physical contact between test subjects and rats that are subsequently placed in the corrals. After a 1 hr habituation period to the plastic corrals inserted in the chamber, the test animal was simultaneously exposed to the same juvenile that it encountered the day before and to a novel juvenile of a different strain for 5 min. In this encounter, each of the juvenile rats was confined to one of the two transparent plastic corrals. The duration of investigatory behavior of the adult test subject toward each juvenile was measured as previously described (*Figure 2B*) (*Gur et al., 2014*). The short-term social memory test, was carried out similarly to the long-term social discrimination test except for three differences: (1) the time between the first and second encounter was 30 min (compared to 24 hr in the long-term social discrimination test), (2) the test rat was habituated to the corrals for 1 hr before the first encounter (instead of the second encounter) to prevent the novelty of the environment from interfering with the expression of social behaviors, and (3) The first encounter was conducted for 5 min while the juvenile was held in one of the corrals (*Figure 2A*). Investigation time was scored by an experienced observer who was blind to genotype.

## Object location memory

Test rats (9–11 week old) were placed for 1 hr in a chamber (60 × 40 × 20 cm), which contained two identical LEGOs. The locations of the LEGOs were fixed by being firmly magnetized to the chamber. The same animals were placed 24 hr later in the same chamber for 1 hr for habituation and then exposed for 5 min to the same LEGOs they encountered the day before. However, during this session the location of one of the LEGOs was different than during the first session. The duration of investigatory behavior toward each of the LEGOs was measured, including any contact between the subject's nose and the object or any intended touch of the object by the adult test subject (*Figure 2C*). These subject-object interactions were scored by an experienced observer who was blind to genotype.

## Contextual fear memory

The contextual fear memory test was carried out as previously described (*Kritman and Maroun, 2013*). Briefly, adult rats (9–11 week old) were placed in fear conditioning chambers (Panlab, Harvard Apparatus; Holliston, MA), which are outfitted with electrified floor-grids, black methacrylate walls, and a transparent front door. On conditioning days, the rats received three 0.5 s foot-shocks delivered through the grid at an intensity of 0.8 mA. The three shocks were administered 90, 210, and 330 s after the subjects were placed in the chamber. After delivery of the third shock, freezing behavior was measured for 2 min to evaluate the success of the fear conditioning protocol. 24 hr after fear conditioning, the rats were placed in the same chamber for 5 min without being subjected

to foot-shock. Freezing behavior during a 5 min test period was quantified offline, averaged and presented as a percentage of the total session time spent freezing (*Figure 2D*). Testing and quantification were performed by an examiner blind to subject genotype.

## 5-Choice serial reaction time (5-CSRT) task

Eight-week-old rats were habituated to being handled by the experimenter and were food deprived to achieve ~85% of free feeding weight. 5-CSRT task training occurred in a manner similar to what has previously been described (*Mar et al., 2013*). Briefly, rats were trained to touch the location of an illuminated white square that is presented at 1 of 5 locations on a Bussey-Saksida capacitive touchscreen system (Lafayette Instrument Company; Lafayette, IN, USA). If a capacitive screen touch occurred at the illuminated location during or up to 1 s after stimulus presentation a sugar pellet was delivered in the reward receptacle located across the chamber from the touch screen. Training occurred in stages where the duration of the light stimulus was slowly decreased from 32 to 1 s by halving the stimulus duration across sessions once criterion performance was met (~80% accuracy and less than 20% of trials omitted). 4 WT, 5 Het, and 3 KO rats were unable to reach performance criterion at 1 s and therefore were excluded from analysis of 5-CSRT task performances. Analysis of these data was performed with custom routines written in the R statistical programming environment (*R Development Core Team, 2006*) and the results are presented in *Figure 3A–D* and *Figure 3—figure supplement 1*. Training, testing, and analyses were performed blind to subject genotype.

## Electrophysiology

### Hippocampal slice preparation

Rats were deeply anesthetized with isoflurane and then sacrificed by decapitation. The brain was removed and quickly placed in ice-cold cutting solution containing the following (in mM): 215 sucrose, 20 glucose, 26 $NaHCO_3$, 4 $MgCl_2$, 4 $MgSO_4$, 1.6 $NaH_2PO_4$, 1 $CaCl_2$, and 2.5 KCl. Hippocampi were mounted on an agar block, and transverse slices 400 μm thick were prepared with a DTK-2000 microslicer (Dosaka EM). Slices were placed in a hot water bath at 30°C, in a holding chamber containing 50% cutting solution and 50% artificial CSF (ACSF) recording solution, which contained the following (in mM): 124 NaCl, 26 $NaHCO_3$, 10 glucose, 2.5 KCl, 1 $NaH_2PO_4$, 2.5 $CaCl_2$, and 1.3 $MgSO_4$. After 30 min, the 1:1 solution was switched to ACSF at 30°C and the holding chamber was removed from the water bath. Slices recovered in ACSF at room temperature for at least 1 hr, and then were transferred to a submersion-type, temperature-controlled recording chamber (TC-344B, Warner Instruments) and perfused with ACSF at 2 ml/min using a peristaltic pump (Dynamax RP-1, Rainin). Experiments were performed at 28°C. All solutions were equilibrated for at least 30 min with 95% $O_2$ and 5% $CO_2$, pH 7.4. Conventional field recordings were performed using a Multi-Clamp 700B amplifier (Molecular Devices). For field experiments, a recording electrode fabricated on a two-step micropipette puller (PP-830 or PC-10, Narishige) filled with a solution containing 5 M NaCl. This electrode was placed ~150 μm deep in the stratum radiatum. To elicit synaptic responses, paired, monopolar square-wave voltage or current pulses (100–200 μs pulse width) were delivered through a stimulus isolator (Isoflex, AMPI) connected to a broken tip (~20–40 μm) stimulating patch-type micropipette filled with ACSF. In some experiments, a bipolar stimulating electrode was used. As both stimulations produced comparable results, the results were averaged together in the final analysis. Typically, stimulating electrodes were placed in the middle of CA1 stratum radiatum. Stimulus intensity was adjusted to give comparable magnitude synaptic responses across experiments (~1 mV for extracellular field recordings). Baseline stimulation was delivered at 0.1 Hz, before and after the induction of long-term plasticity. All neurophysiological data were acquired and analyzed blind to genotype.

### Extracellular recordings

Hippocampal slices (350 μm, Cohort 1; 400 μm, cohort 2) were prepared from 4-6 week old (cohort 1) and 6-8 week old (cohort 2) Het and KO rats and their wild-type littermate controls. As previously described, stimulation electrodes were placed in CA3 of stratum radiatum and field excitatory postsynaptic potentials (fEPSPs) were recorded from CA1 of stratum radiatum (*Bozdagi et al., 2000*; *Klein et al., 2015*). Test stimulus intensity was adjusted to obtain fEPSPs with amplitudes that were one-half of the maximal response. The EPSP initial slope (mV/ms) was determined from the average

waveform of four consecutive responses (*Figure 4—figure supplement 1*). Paired-pulse responses were measured with inter-stimulus interval (ISI) of 50 ms, and are expressed as the ratio of the average responses to the second stimulation pulse (FP2) to the first stimulation pulse (FP1) (see *Supplementary file 1*). Long-term potentiation (LTP) was induced by either a high-frequency stimulus (four trains of 100 Hz, 1 s stimulation separated by 5 min, *Figure 4A*) or a single 100 Hz stimulation (*Figure 6—figure supplement 1A*) . (S)−3,5-dihydroxyphenylglycine (DHPG, 50 μM) was used to induce mGluR-dependent LTD (*Figure 4B* and *Supplementary file 1*). Oxytocin (1 μM) (*Tomizawa et al., 2003*; *Lin et al., 2012*) was bath-applied, for 20 min in the baseline as indicated in *Figure 6A* and *Figure 6—figure supplement 1A* and in the figure legends.

### Whole-cell recordings

For mPFC recordings, 300-μm-thick coronal brain slices were prepared. Recording methods were similar to previously published protocols (*Bozdagi et al., 2000*; *Li et al., 2011*; *Wang et al., 2011*). Acute slices were prepared from 4- to 6-week-old *Shank3* Het and KO rats or WT littermates. All recordings were conducted in layer five pyramidal cells in slices kept at 32°C (*Supplementary file 1*).

### In vivo field recordings in anesthetized rats

*Shank3* Het or KO rats and WT littermate controls were anesthetized with urethane (1.5 g/kg, i.p.) and placed in a stereotaxic frame and prepared for recordings as previously described (*Bozdagi et al., 2007*). For mPFC recordings, unilateral craniotomies were drilled over mPFC and glass recording microelectrodes were positioned according to stereotaxic coordinates (AP = 3 mm, ML = 0.6 mm, DV = 5 mm). To stimulate hippocampal projections, a stimulation electrode was placed in hippocampal CA1 (AP = 6.3 mm, ML = 5.2 mm lateral, DV = 8 mm). After baseline responses were recorded for 30 min, LTP was induced using 4 trains of 100 Hz stimulation applied every 5 min (*Figure 4C*). Oxytocin (2 ng/4 μl with a rate of 1 μl/min) was injected ICV (AP = −0.3 mm, ML = 1.3 mm, DV = −3.5 mm) using a precision syringe (Hamilton), 0–5 min prior to HFS (*Figure 6B*). All neurophysiological data were acquired and analyzed blind to genotype.

## Surgical implantation of guide cannula

For the SRM experiments, a guide cannula was implanted in naïve rats at 8.5–9.5 weeks of age. For the 5-CSRT task, surgery occurred after baseline performance was measured for 10 sessions (after criterion performance was first met on with a 1 s cue duration). For surgery, rats were deeply anesthetized by subcutaneous injection of 10% ketamine (0.09 ml/100 g) and Dormitor (0.05 ml/100 g) or with isoflurane. Anesthetized animals were fixed in a stereotaxic apparatus with the head flat, and a small hole was drilled in the skull to facilitate implantation of guide cannula. The cannula was inserted 0.1 mm (SRM task) or 1.25 mm (5-CSRT task) above the left lateral ventricle. The coordinates were AP = −1 mm, ML = 1.5 mm, DV = −3.6 mm (SRM experiments) or AP = −0.75 mm, ML = 1.75 mm, DV = −4 mm (5-CSRT task) both targeting the left lateral ventricle. Cannula were secured in position with acrylic dental cement and stabilized by at least two bone screws. A dummy cannula was placed in the guide cannula to prevent the tube from clogging. Antibiotics (amoxicilin 15%, 0.07 ml/100 g) and a pain killer (Calmagin 0.03 ml/100 g) were administered directly before and immediately following surgery and one day post-surgery. Animals were allowed to recover for at least 7 days before experimentation began. Post-surgery, 5-CSRT task rats were retrained to stable performance before being subjected to microinjections. In the oxytocin 5-CSRT experiments, two implanted KOs, 1 Het and WT were not subjected to microinjections because they failed to achive pre-surgery performance levels. In the long-term SRM task, 1 WT and 2 KOs lost their caps, 1 Het and 1 KO died during surgery, and 1 Het was extremely stressed and thus was not subjected to oxytocin injection. All experiments were run and analyzed blind to genotype.

## Oxytocin administration

The injectate for intracranial microinjections was prepared fresh daily by dissolving oxytocin (American Peptide Company) in 0.9% saline to achieve a final concentration of 250 nM (0.25 ng/μl) (*Benelli et al., 1995*).

On test days, the fluid line and the microinjector were filled with mineral oil before drawing up the drug or saline control. Prior to microinjection, animals were gently restrained while the dummy

cannula was removed and the microinjectors loaded with injectate were placed into the animal's guide cannula. The injectate was administered at a rate of 1 μl/min for 4 min (in total 1 ng of oxytocin was injected). Concentration was chosen based on published studies of the effect of ICV oxytocin injection on behavior (*Ferguson et al., 2000*; *Gur et al., 2014*). After injection, the microinjector remained in the guide cannula for 1 min to allow for diffusion. In the SRM task, microinjections occurred 10–20 min before the first social encounter (*Figure 5A* and *Figure 5—figure supplement 1A*). In the 5-CSRT task, animals were immediately placed in the operant chamber after oxytocin administration and testing started 10 min after (*Figure 5B* and *Figure 5—figure supplement 1B*). The order of saline and oxytocin injections was randomized across rats.

## Histology
At the end of each of experiment rats were sacrificed with an overdose of isoflurane, brains were removed and placed in 4% paraformaldehyde overnight, and 200-μm-thick sections were prepared. Correct positioning of the cannula over the left lateral ventricles was confirmed visually in each rat.

## Statistical analysis
SPSS (SPSS 19; IBM) or the R environment were used for statistical analyses. Parametric approaches such as t-tests or analysis of variance were used if data was found to be normally distributed by the Kolmogorov-Smirnov test. When the condition of normality was not met, non-parametric approaches for statistical comparisons were used such as Kruskal-Wallis or the Wilcoxon signed rank test. A threshold of $p < 0.05$ was used to test statistical hypotheses. A detailed description of the results of all analyses, including the test used, sample size, mean or median, SEM or SD, and p values can be found in *Supplementary file 1*.

## Acknowledgements
This work was supported by the National Institutes of Health/National Institute of Mental Health grants R01 MH093725 and R01 MH101584 to JDB, R01 MH081935 and R01 DA017392 to PEC, F31 NS073200 and T32 GM007288 to MEK, the Seaver Foundation to HHN, the Human Frontier Science Program (HFSP) RGP0019/2015 to HHN, JDB and SW, Autism Speaks 8410 to JDB, and by the Israel Science Foundation (SFN) grant 1350/12 to SW. In addition, this study was supported by a generous gift from William G. Gibson and Paulina Rychenkova, PhD. We thank Dr. Mouna Maroun for her advice on the fear conditioning test, Dr. Jill Silverman for reviewing the manuscript, and Ms. Idil Ozturk, Ms. Ayesha Islam, and Ms. Deniz Bingul for their assistance in behavioral scoring.

## Additional information

### Funding

| Funder | Grant reference number | Author |
| --- | --- | --- |
| Human Frontier Science Program | RGP0019/2015 | Hala Harony-Nicolas Shlomo Wagner Joseph D Buxbaum |
| National Institute of Mental Health | F31-NS073200 | Matthew E Klein |
| National Institute of Mental Health | T32-GM007288 | Matthew E Klein |
| National Institute of Mental Health | R01-MH081935 | Pablo E Castillo |
| National Institute of Mental Health | R01-DA17392 | Pablo E Castillo |
| Israel Science Foundation | 1350/12 | Shlomo Wagner |
| National Institute of Mental Health | R01-MH093725 | Joseph D Buxbaum |
| National Institute of Mental | R01-MH101584 | Joseph D Buxbaum |

| | | |
|---|---|---|
| Health | | |
| Autism Speaks | 8410 | Joseph D Buxbaum |

The funders had no role in study design, data collection and interpretation, or the decision to submit the work for publication.

## Author contributions

HH-N, Conceptualization, Data curation, Formal analysis, Funding acquisition, Validation, Investigation, Methodology, Writing—original draft, Project administration; MK, Conceptualization, Data curation, Formal analysis, Validation, Investigation, Methodology, Writing—original draft, Writing—review and editing; JdH, Conceptualization, Data curation, Formal analysis, Investigation, Methodology, Writing—original draft, Writing—review and editing; MEK, Data curation, Formal analysis, Investigation, Methodology, Writing—review and editing; OB-G, Conceptualization, Data curation, Formal analysis, Methodology, Writing—review and editing; MR, SS, Data curation, Writing—review and editing; NPD, Data curation, Formal analysis, Methodology, Writing—review and editing; PEC, Supervision, Funding acquisition, Writing—review and editing; PRH, Conceptualization, Supervision, Funding acquisition, Investigation, Writing—review and editing; MLS, Conceptualization, Resources, Supervision, Investigation, Methodology, Writing—original draft, Writing—review and editing; MGB, Conceptualization, Resources, Data curation, Formal analysis, Supervision, Investigation, Methodology, Writing—original draft, Writing—review and editing; SW, Conceptualization, Resources, Formal analysis, Funding acquisition, Investigation, Writing—original draft, Writing—review and editing; JDB, Conceptualization, Resources, Formal analysis, Supervision, Funding acquisition, Validation, Investigation, Methodology, Writing—original draft, Writing—review and editing

## Author ORCIDs

Hala Harony-Nicolas, http://orcid.org/0000-0001-8283-9093
Shlomo Wagner, http://orcid.org/0000-0002-7618-0752
Joseph D Buxbaum, http://orcid.org/0000-0001-8898-8313

## Ethics

Animal experimentation: All animal procedures were approved by the Institutional Animal Care and Use Committees at the Icahn School of Medicine at Mount Sinai (protocol number: LA12-00270), at the University of Haifa (Protocol number: 194-10), and at the Albert Einstein College of Medicine (Protocol number: 20130506)

## Additional files

**Supplementary files**

• Supplementary file 1. Summary of all experimental and statistical results.

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
