## [Decision Letter]

[Editors’ note: this article was originally rejected after discussions between the reviewers, but the authors were invited to resubmit after an appeal against the decision.]

Thank you for submitting your work entitled "Oxytocin improves social and attentional deficits in a novel genetically modified rat model of autism" for consideration by *eLife*. Your article has been favorably evaluated by a Senior Editor and three reviewers, one of whom, Peggy Mason (Reviewer #1), is a member of our Board of Reviewing Editors, and another one is Karen Bales (Reviewer #2).

Our decision has been reached after consultation between the reviewers. Based on these discussions and the individual reviews below, we regret to inform you that your work will not be considered further for publication in *eLife*.

Obtaining mechanistic insight into autism and related conditions through the analysis of rodent models of this disease is of major importance. Thus, the topic of this manuscript, that the *Shank3*-KO rat is a model of PMS, is of great interest and timeliness. Unfortunately, the reviewers were not convinced that the *Shank3* rat has face validity with PMS. The deficits that were explored appear to be in memory and possibly perception more so than in social behavior. Additional concerns that we hope will aid the authors are detailed in the reviews appended below.

*Reviewer #1:*

This manuscript shows that oxytocin ameliorates several of the differences between *Shank3*-Het and KO rats and WT rats. Several issues concern this reviewer.

The short-term vs. long-term SD tasks are termed in a confusing or even misleading fashion. In the short-term SD, rats interact for 5 mins and then are tested 30 mins later. In the long-term SD, they interact for 60 mins and then are tested a day later. Which of the changes is relevant – the time of exposure or the interval before testing? And which change – both of which lengthen – does the short vs. long refer to? This is not a trivial point as the Het/KO rats are fine at the short-term and not at the long-term. And this is interpreted as a difference in memory. It could be due to a difference in exploration or perception.

The perception vs. memory difference is potentially important. The fact that they do better after a 5 s light CSRT than a 1 s one suggests that there may be a deficit in perception. 1 s is too short for them to interpret correctly. In the last paragraph of the subsection “Social memory and attentional deficits in *Shank3*-deficient rats”, it is unclear how this result is interpreted as it is lumped with an unrelated finding that speaks to the motivation for food reward.

The object location task is not exactly parallel to the SD task. Location rather than identity is tested. Why? Why not make the tasks entirely parallel?

Figure 4. Is there enough power (6 rats per genotype) to say that there is no difference? Of note the rank order is WT> Het> KO for both NMDA and AMPA. In contrast it looks as though the n per group for LTP is as much as 60 (6 rats and 10 slices /rat). The actual number of data points in 4B should be stated.

How are the numbers of rats tested determined? E.g. for the oxt studies with CSRT, 15 HETs and 8 KOs were studied. Of concern was that the numbers were not pre-set.

*Reviewer #2:*

This paper behaviorally and neurobiologically characterizes the *Shank3* rat, as a model for autism, and examines the effects of oxytocin administration on the behaviors. In addition, it examines LTP in hippocampal slices in the *Shank3* rat and shows that oxytocin restores normal LTP in vitro. This paper makes a large contribution to the literature on oxytocin through its thorough behavioral characterization and also in that it showed a neural mechanism for oxytocin to ameliorate behavioral deficits.

I have two large concerns:

Whereas the main therapeutic use for oxytocin in humans is via intranasal administration, in this study it was via intracerebroventricular administration. This is not clear unless you look at the methods. Please modify the Results/Discussion to indicate this mode of administration, and that results might differ from intranasal administration.

Also:

The Methods say that the EPM was performed (subsection “Elevated plus maze”), but I could not find these results in the text.

*Reviewer #3:*

In this manuscript Harony-Nicolas et al. describe the successful generation of *Shank3*-deficient rats, modeling the Phelan-McDermid syndrome (PMS) in humans. They report that these rats show several behavioral impairments related to social memory and attention that are associated with PMS, and then go on to show that electrophysiological phenotypes in the hippocampus and prefrontal cortex are present in these rats that may serve as potential mechanisms for the observed behavioral deficits. These phenotypes were reversed by in vitro or in vivo administration of oxytocin. These novel findings are of high interest to the field as using rats as model organisms for ASD could be beneficial for translating potential treatment options. However, there is a disconnect between the presented findings and the authors' claims that these rats serve as an animal model of ASD – these rats do not show any of the core behavioral symptoms of ASD (repetitive behavior, social interaction deficits, etc.). The phenotypes are more related to learning and memory, which are affected in patients with *Shank3* mutations. Thus, the title and Abstract are misleading. Overall, a more transparent and accurate assessment of the Results is necessary, rather than trying to fit this model into the "ASD box. " In fact, it is quite interesting that these animals do not mirror the mouse models, and do not have behavioral manifestations expected in a model of ASD, but rather a model of ID.

1) The title of the paper suggests that these rats display "social deficits" – this should be changed to "memory deficits" since these rats do not appear to have a generalized social deficit. This is a general theme throughout the paper that needs to be seriously addressed

2) Many of the behavioral phenotypes exhibited by several types of *Shank3*-deficient mouse models of ASD, for example elevated anxiety, repetitive behavior, reduced social interest (e.g. Peca et al., 2011 Nature; Mei et al., 2016 Nature; Wang et al., 2016 Nat. comm.) are not seen in these *Shank3*-deficient rats. Do these rats develop compensatory mechanisms that protect from these behavioral phenotypes associated with ASD? Does the function of corticostriatal circuits remain intact?

3) The presented dataset suggests that there is a specific impairment in hippocampal synaptic plasticity, and not areas involved in social cognition, such as frontal lobe – at the same time object location memory remains intact in the rats. Are other hippocampal plasticity-dependent tasks impaired in these mice? Given the uncertainty as to which circuits are involved, this needs to be presented clearly.

4) We have no real insight into mechanisms, which is the reason to use the model system. What are the mechanisms of impaired LTP in the hippocampus? Are there postsynaptic components, as expected from *Shank3*'s role as a postsynaptic protein?

5) The recovery of behavioral impairments by i.c.v. oxytocin is an exciting finding. However, the authors have not addressed which receptor type, expressed in which brain region(s), is responsible. Oxytocin can activate both vasopressin and oxytocin receptors, and these receptors are widely expressed throughout the brain. Since the authors have shown that the synaptic plasticity in the hippocampus is impaired whereas it remains intact in the PFC, oxytocin is presumably acting on the hippocampus to recover the attentional deficits and memory. Is there any experimental evidence to support this?

Again, social memory and attention while sometimes disrupted in ASD are not core features of ASD. The hippocampal deficits that are suggested here speak more to learning and memory and therefore, ID, in humans.

[Editors’ note: what now follows is the decision letter after the authors submitted for further consideration.]

Thank you for submitting your article "Oxytocin improves behavioral and electrophysiological deficits in a novel *Shank3*-deficient rat" for consideration by *eLife*. Your article has been favorably evaluated by a Senior Editor and two reviewers, one of whom, Peggy Mason (Reviewer #1), is a member of our Board of Reviewing Editors, and another one is Karen Bales (Reviewer #2).

The reviewers have discussed the reviews with one another and the Reviewing Editor has drafted this decision to help you prepare a revised submission.

This manuscript reports a valuable description of a *Shank3* mutant rat, its cognitive deficits and rescue of the deficits tested by ICV oxytocin.

The authors are asked to address the sample sizes one last time. The authors say that the samples were largely constrained by the 25-50-25 inheritance pattern of WT-Hets-KOs. Yet the samples don't show this 25-50-25 distribution. They are very close to 1:1:1 overall. In the early tests – where mortality does not figure and where the distribution should be closer to Mendelian – the rats are at 9-15-12 which is a ratio of 0.6-1-0.8. However, in the core social memory tests (conducted later) the distributions are 12-13-9, 22-20-11, 9-17-14, 31-37-25, averaging 0.9-1-0.7. Then in the oxytocin rescue experiments (which are if anything more time consuming and subject to loss), the ratio is again very close to 1:1:1.

These ratios close to one are not easy to understand and are not explained by the authors' letter. The reason that this is important is because of the need to set sample size targets for behavioral tests prior to conducting the tests in order to avoid unconscious bias to creep into the experiment.

It might be of help to discuss how many animals in each pool and of each type were lost to surgery or some other fatality.

Was the person conducting the e-phys blind to the rat's genotype? And the person that analyzed the e-phys’ results?

Given the history of this submission, we must reach a final decision on your next and last version.

*Reviewer #2:*

The authors have addressed all of my comments and I have no further concerns.

---

## [Author Response]

[Editors’ note: the author responses to the first round of peer review follow.]

*[…] Reviewer #1:*

*This manuscript shows that oxytocin ameliorates several of the differences between Shank3-Het and KO rats and WT rats. Several issues concern this reviewer.*

*The short-term vs. long-term SD tasks are termed in a confusing or even misleading fashion. In the short-term SD, rats interact for 5 mins and then are tested 30 mins later. In the long-term SD, they interact for 60 mins and then are tested a day later. Which of the changes is relevant – the time of exposure or the interval before testing? And which change – both of which lengthen – does the short vs. long refer to? This is not a trivial point as the Het/KO rats are fine at the short-term and not at the long-term. And this is interpreted as a difference in memory. It could be due to a difference in exploration or perception.*

We thank the reviewer for raising this point. To avoid further confusion, we have now clarified in the Results section that the long- and short-term refer to the interval between exposures and not the exposure time (subsection “Social memory and attentional deficits in *Shank3*-deficient rats”, second paragraph). This is in addition to the diagrams that we included in the original manuscript (Figure 2), which outlined the method schematically (including showing the timing of the exposure and the interval), and the Methods section that detailed both paradigms.

We understand the reviewer’s concern and agree that, under certain circumstances (that we don't think apply here) the difference in the time spent with the social stimuli may introduce confounds that relate to exploration or perception. However, this would only be of concern if the rats showed deficits in their short-term social recognition memory (with the short interaction time) while showing intact long-term social recognition memory (with the longer interaction time). The opposite is true. The *Shank3* HET and KO rats show intact short-term memory on the short-term social discrimination test even with a short exploration time, indicating that their social exploration or perception is not affected. And they show an impaired behavior on the long-term social discrimination test thus suggesting that their long-term memory is affected, even when given enough time (1 hour) for exploration. Furthermore, the possibility of differences in exploration is addressed by the similarity in total exploration time between the genotypes and in their social preference. We have now added and discussed these points in the Results section (subsection “Social memory and attentional deficits in *Shank3*-deficient rats”, second paragraph).

Finally, it is noteworthy to mention that it is a common procedure in memory studies that a weaker stimulus is used to induce short-term memory, while a stronger stimulus are used (and often required)to produce long-term memory. As an example is the low and high doses of LiCl used in the conditioned taste aversion paradigm (Houpt&Berlin 1999, PMID:10355522). This, in fact, correlates with LTP procedures where short-term LTP can be achieved with one train of tetanic stimulation while long- term LTP requires several such trains. The idea behind using distinct procedures in our experiments is to apply a minimal exposure that generates a stable memory, so if indeed any of the genotypes exhibit impaired social memory it will be easily detected on the background of such a minimal procedure. For short-term memory, exposure of 5 minutes is enough, and hence used here, while a much longer exposure (1 h in our hands) is needed for long-term SRM (in rats).

*The perception vs. memory difference is potentially important. The fact that they do better after a 5 s light CSRT than a 1 s one suggests that there may be a deficit in perception. 1 s is too short for them to interpret correctly. In the last paragraph of the subsection “Social memory and attentional deficits in Shank3-deficient rats”, it is unclear how this result is interpreted as it is lumped with an unrelated finding that speaks to the motivation for food reward.*

We understand the concern of the reviewer and agree that changes in accuracy can be conceived as reflecting deficits in sensory perception, specifically those related to basic visual sensory function. However, it is important to note that the pattern of deficit we observe, and which is evoked by only changing the stimulus duration, is commonly interpreted as reflecting an attentional deficit, while deficits observed by changing the physical qualities of the light(brightness or illuminance) are those that reflect perceptual differences (Reviewed by Robbins TW, 2002, PMID: 12373437). In our experiment we used the same light stimulus in all sessions and found that the rats have no problem in seeing and responding to the light, implying that they have no issues with their sensory perception. We also did a decreased brightness challenge in one of the rat cohorts and did not observe any effect on performance, which further argues against the observed primary deficit being due to perceptual problems. Only when the duration of the light was shortened, the deficits were evoked and the rats had difficulties in detecting a more briefly presented light, indicating that the *Shank3* Het and KO rats have difficulty in vigilance or spatial attention. We have now clarified this point in the Results section (subsection “Social memory and attentional deficits in *Shank3*-deficient rats”, last paragraph).

We have edited the sentence that the reviewer refers to (in the original file) to clarify this point, and want to note that the sentence that follows (“These deficits were not due to decreased motivation for food reward, because latency to collect reward after a correct response was similar among genotypes (Figure 2—figure supplement 1), as was task performance when light cues were of a longer duration” is not unrelated, but rather is important because the observed attentional findings could be also related to motivational problem, which we rule out by examining the latency of the rats to collect reward, which we found to be unaffected in the *Shank3*-deficient rats.

Finally, as a side comment, we want to note that the reviewer’s comments: “The fact that they do better after a 5 s light CSRT than a 1 s one suggests that there may be a deficit in perception”, if thought about more generally, supports the second point we raised for the previous comment and emphasizes the notion that perception would be an issue if the rats perform worse on shorter stimuli vs. longer ones on the social recognition memory test and not vice versa.

*The object location task is not exactly parallel to the SD task. Location rather than identity is tested. Why? Why not make the tasks entirely parallel?*

The idea behind testing the *Shank3* rats on the object location memory and the contextual fear conditioning tests was not to examine a paradigm that is exactly parallel to social discrimination. Instead, the aim was to examine whether the observed deficits in social recognition memory are specific to social stimuli or that the *Shank3* rats have a general memory deficit. Obviously, the first choice one would think about to test non- social memory is the novel object recognition test, given the fact that it technicallyseems to be the best to parallel the social discrimination test that we applied. However, the reason we focused on the object location memory and contextual fear conditioning tests in our study is because social recognition memory is well known to be hippocampal- dependent and our findings demonstrate that the *Shank3* rat model has hippocampal LTP deficits. Therefore, we aimed to examine other hippocampal-dependent memory paradigms. Both the object location and the fear conditioning tests are known to be hippocampal dependent, while the dependence of the object location memory test on the hippocampus is not clear (see for example Gaskin et al. 2003, PMID: 14750658). Since the object location memory and the contextual fear conditioning tests differ in their mode (the fear-conditioning task is a reinforcement-driven task while the object location memory test depends on the animal’s natural exploratory tendency when introduced in a new stimulus or stimulating environment), we decided to include both. We have further clarified this point in the Results section (subsection “Social memory and attentional deficits in *Shank3*-deficient rats”, third paragraph).

*Figure 4. Is there enough power (6 rats per genotype) to say that there is no difference? Of note the rank order is WT> Het> KO for both NMDA and AMPA. In contrast it looks as though the n per group for LTP is as much as 60 (6 rats and 10 slices /rat). The actual number of data points in 4B should be stated.*

We thank the reviewer for raising this point and agree that we are underpowered to exclude a real change in I/O function for NMDA. We estimate that we need 15 data points to say that reliably. We have therefore removed this figure and repeated the I/O function assessment in an independent set of experiments on WT and KO rats, focusing on overall I/O and using larger n (n=5 rats per genotype/3 slices per rat). This data is now presented as Figure 4—figure supplement 1, and shows no significant differences in I/O function between genotypes. Understanding subtle impact of *Shank3* deficiency on multiple aspects of synaptic function (including AMPA and NMDA receptors) is an important study that will require multiple different approaches and analyses. Altogether, the data suggest that change in I/O function is not likely to explain the clear LTP changes.

We apologize for the typographical error in the number of animals used for LTP recording. The correct numbers were n=6 rats/1-2 slices per rat. Those numbers have been corrected in the legend of the figure and in the supplementary file ([Supplementary-material SD1-data]).

*How are the numbers of rats tested determined? E.g. for the oxt studies with CSRT, 15 Hets and 8 KOs were studied. Of concern was that the numbers were not pre-set.*

The sample size was chosen based on the standards in the field and using on-line sample size/power calculation tools. Specifically, we followed the recommendation for the “Minimum Numbers of Mice” discussed by Professor Jacqueline N. Crawley, in her book entitled “What’s Wrong With My Mouse”, Second Edition and applied the “biomath” online tool; (http://biomath.info/power/) for targeted calculations. Following the book’s recommendation, we determined our minimal N following Figure 1 (contributed by Professor Douglas Wahlsten) in the above-mentioned book, which describes the number of animal/per group needed to obtain specific statistical power. In general, our N for behavioral studies ranged between 9-22, therefore we were 60-80% powered to detect large effect differences (δ=1 – 1.5). As noted, we also calculated the numbers for each experiment based on its design using the “biomath” online tool. For example, for the social discrimination test, we applied a paired t-test between the novel and familiar stimuli to determine SRM (this was also applied for the Oxt treatment experiments). Since in normal conditions rats spend on average at least 60% of their total investigation time exploring the novel stimulus, with SD of 12%, a sample size of n=6 is calculated for 80% power and confidence interval of 5%. However, we always try to double this sample size so we aim for 12 animals per group, beyond the predicted sample size. This was important because, in some cases, we lost some animals during the experiment, leaving us with 7-8 animals, which is still beyond the predicted sample size. Information about the exact N and statistical method used for each experiment were detailed in the supplementary file ([Supplementary-material SD1-data]).

We understand the question of the reviewer regarding the variability of the N between the different genotypes. In all our studies we use offspring of Het x Het breeding. This is to be able to study littermates, which is optimal when carrying out behavioral studies. The fact that the numbers varied between the genotypes is hence primarily because HET x HET breeding produces a distribution of 25% WTs, 25% KOs and 50% HETs. Therefore, for each WT and KO rats we have two HET littermates that we could use and include in our data, when available, without invalidating our experiment and analysis. The 5-CSRTT requires at least two months of training after which we do surgeries to inject the OXT (in the case of the 5-CSRTT-OXT experiment). During this period and the surgeries, there is always a chance of losing one or two animals, which cannot be replaced, together with its littermates, before 4 additional months (breeding, reaching the age of testing and then training). Therefore, we tried to include as many of the available rats as possible, given the limited number of testing chambers, even if this means that we will not start or end with precisely equal numbers of animals.

*Reviewer #2:*

*This paper behaviorally and neurobiologically characterizes the Shank3 rat, as a model for autism, and examines the effects of oxytocin administration on the behaviors. In addition, it examines LTP in hippocampal slices in the Shank3 rat and shows that oxytocin restores normal LTP in vitro. This paper makes a large contribution to the literature on oxytocin through its thorough behavioral characterization and also in that it showed a neural mechanism for oxytocin to ameliorate behavioral deficits.*

*I have two large concerns:*

*Whereas the main therapeutic use for oxytocin in humans is via intranasal administration, in this study it was via intracerebroventricular administration. This is not clear unless you look at the methods. Please modify the Results/Discussion to indicate this mode of administration, and that results might differ from intranasal administration.*

We thank the reviewer for raising this point and agree that this is an important issue that should be discussed and clarified in the manuscript. We agree with the reviewer that it is totally possible that when ICV injected, the effect of oxytocin may be different than when it is intranasally administered. This is mainly due to the fact that the percentage of oxytocin that reaches the brain following intranasal administration (using tolerable concentrations) is extremely small making the brain oxytocin concentration following intranasal administration incomparable to those detected following direct ICV injection into the brain. This reality implies that the delivery approach of oxytocin to the brain should be improved to allow better comparison between direct and indirect administrations in animal studies and to optimize treatment efficacy in human subjects. We have now added a paragraph to address this issue in the Discussion section (eighth paragraph).

*Also:*

*The Methods say that the EPM was performed (subsection “Elevated plus maze”), but I could not find these results in the text.*

We thank the reviewer for pointing out that EPM results were missing in the text. The approach was indeed discussed in the Methods section, and the results and statistical analysis were included in the supplementary file, but not in the main text. We have now added this information to the Results section (subsection “Production and developmental phenotyping of the *Shank3*-deficient rat model”, last paragraph).

*Reviewer #3:*

*In this manuscript Harony-Nicolas et al. describe the successful generation of Shank3-deficient rats, modeling the Phelan-McDermid syndrome (PMS) in humans. They report that these rats show several behavioral impairments related to social memory and attention that are associated with PMS, and then go on to show that electrophysiological phenotypes in the hippocampus and prefrontal cortex are present in these rats that may serve as potential mechanisms for the observed behavioral deficits. These phenotypes were reversed by in vitro or in vivo administration of oxytocin. These novel findings are of high interest to the field as using rats as model organisms for ASD could be beneficial for translating potential treatment options. However, there is a disconnect between the presented findings and the authors' claims that these rats serve as an animal model of ASD – these rats do not show any of the core behavioral symptoms of ASD (repetitive behavior, social interaction deficits, etc.). The phenotypes are more related to learning and memory, which are affected in patients with Shank3 mutations. Thus, the title and Abstract are misleading. Overall, a more transparent and accurate assessment of the Results is necessary, rather than trying to fit this model into the "ASD box. " In fact, it is quite interesting that these animals do not mirror the mouse models, and do not have behavioral manifestations expected in a model of ASD, but rather a model of ID.*

We appreciate and respect the reviewer’s point of view and agree that the phenotypes observed in the *Shank3* rat model may relate more to those affected in patients with PMS and have changed the title and text accordingly.However, we still believe that the *Shank3* rat is a valid model for important aspects of ASD, as mutations in this gene are associated with such high risk for ASD. As we and others have shown, not all patients with PMS have ASD, so there is no reason to expect that a specific rodent model will show ASD (see Soorya et al. 2013, PMID23758760). We added a paragraph that discusses this issue in the Discussion section (second paragraph).

But the broader point is that it has become quite clear that there are many ways to have a deficit in reciprocal social interactions and hence meet criteria in ASD, that have little to do with social aversion or lack of social approach. The simplest example is William Syndrome, known for hypersociability, where some 50 percent of individuals meet criteria for autism. They show excessive social approach but they are not able to carry out ‘typical’ reciprocal social interactions.

In the specific case of PMS, in our recent functional neuroimaging study we show quite clearly that differential fMRI changes in response to social versus non-social sounds are *preserved* in PMS (see Wang et al. 2016, PMID: 26909118), a difference that is absent in other studies of idiopathic autism. In addition, we show that social orientating is preserved in PMS, again, something reduced in studies of idiopathic autism. For this reason, others and we conclude that PMS-associated ASD is *not* associated with a social indifference or social aversion, in contrast to the more typical conception of ASD. In addition, we conclude that the primary, and most significant and pervasive deficits in PMS are cognitive and attentional, rather than social. And there is little doubt that there will be many subtypes of autism with preserved social interest. Hence we would argue that this is a good model for PMS and a reasonable model for some subtypes of ASD. However, as noted above, we have made the requested changes (Abstract; Introduction; Discussion, second paragraph).

*1) The title of the paper suggests that these rats display "social deficits" – this should be changed to "memory deficits" since these rats do not appear to have a generalized social deficit. This is a general theme throughout the paper that needs to be seriously addressed*

We agree with the reviewer and have now changed the titleto better reflect the findings. However, we did not use the term “memory deficits” as suggested by reviewer. This is because we believe this would be misleading given the fact that the *Shank3*- deficient rats showed no deficit on two non-social long-term memory paradigms, the object location memory and contextual fear conditioning tests, and that their deficits are only on tasks that implicate social-dependent memory. Therefore, in the same way that the reviewer emphasizes that the social deficits cannot be generalized, we believe that the observed memory (social memory) deficits cannot be generalized as well.

*2) Many of the behavioral phenotypes exhibited by several types of Shank3-deficient mouse models of ASD, for example elevated anxiety, repetitive behavior, reduced social interest (e.g. Peca et al., 2011 Nature; Mei et al., 2016 Nature; Wang et al., 2016 Nat. comm.) are not seen in these Shank3-deficient rats. Do these rats develop compensatory mechanisms that protect from these behavioral phenotypes associated with ASD? Does the function of corticostriatal circuits remain intact?*

We agree with the reviewer that the *Shank3* rat model does not display all of the phenotypes observed in various *Shank3*-mouse models, and we thank the reviewer for raising this issue, and helping us realize the importance of discussing it in the paper. We have now included a brief discussion about this issue in the Discussion section (third paragraph).

Finding discrepancies between the rat and mouse models is not totally surprising since such discrepancies have been already reported even between Shank3 mouse models. We have previously summarized these discrepancies in detail (Harony-Nicolas et al., 2015; PMID: 26350728), including discrepancies in reported alterations in motor function, anxiety levels, social preference, interest for social novelty, and ultrasonic vocalizations, which exemplify how difficult it is to draw a unifying phenotype when dealing with different models even of the same species.

These discrepancies within the mice models and between the mice and rat model could be attributed to:

1) The fact that different models have mutations/deletions in different parts of the Shank3 gene. Although all mutations lead to the deletion of the longest *Shank3* isoform, each leaves behind other shorter isoforms. The role of these shorter *Shank3* isoforms is not clear. They can potentially play a compensatory and protective role or, contrary; increase the deleterious effect of the introduced mutation leading to the manifestation of different phenotypes in each model.

2) The use of different behavioral paradigms to assess what is identified as the same behavior.

3) The age of the tested animal on each paradigm.

4) The sex of the tested animals.

5) Genetic differences in background strains used in the mice.

Some of the findings in mice are made in homozygous but not heterozygous animals. Moreover, mice and rats are separated by 12-25 million years in evolution and differ in complex behaviors, thus generating different outcomes in behavioral paradigms. Whether we want to relate these different outcomes to compensatory mechanisms or to an added complexity in cognition and behavior, it is an open argument.

Finally, based on our findings showing that the *Shank3*-deficient rats do not exhibit motor deficits or show any impulsive behavior on the 5-CSRT test, we did not have a strong rationale to focus on the corticostriatal circuits in the current study. We chose to study the plasticity in the hippocampus and the hippocampal-PFC circuit because of their *direct* relevance to the observed attentional and social recognition memory deficits. However, one cannot exclude the possibility that other brain circuits are affected by the *Shank3* mutation, especially corticostriatal circuits, which have been shown to be impaired by *Shank3* mutation in the mouse. Therefore, we agree with the reviewer that the integrity of other circuits should be examined as well in the frame of future studies.

*3) The presented dataset suggests that there is a specific impairment in hippocampal synaptic plasticity, and not areas involved in social cognition, such as frontal lobe – at the same time object location memory remains intact in the rats. Are other hippocampal plasticity-dependent tasks impaired in these mice? Given the uncertainty as to which circuits are involved, this needs to be presented clearly.*

As we have previously noted in our response to comment 3 of reviewer #1 and to our response to this reviewer for comment 1, we have examined two well-known hippocampal-dependent long-term memory paradigms, the object location memory and contextual fear conditioning tests. None of these tests showed a significant difference between genotypes. The fact that socialrecognition memory, which is well known to be hippocampal-dependent, was affected in the *Shank3* rat model, while two other non- socialhippocampal-dependent memories were not, suggests that (1) the memory deficits are social specific and (2) social recognition memory is a more "frangible" type of memory that implicates more complex cues as compared to the non-social memory tested by us. Therefore, when the hippocampal circuitry is even partially impaired, the social recognition memory may be more vulnerable and more affected than other types of memory. Another possibility is that the PFC impairment is the most severe in our study and that this region is needed only for social memory.

*4) We have no real insight into mechanisms, which is the reason to use the model system. What are the mechanisms of impaired LTP in the hippocampus? Are there postsynaptic components, as expected from Shank3's role as a postsynaptic protein?*

We indeed use animal models to gain insight into underlying mechanism, and we do that with the ultimate goal of using them for drug discovery and for the evaluation of novel and existing (FDA approved) drugs. While understanding the mechanisms underlying the LTP deficits is highly important and would shed light on affected mechanisms or pathways that can be of interest for drug discovery, the focus of the current study was on producing, validating and studying a novel rat model and to see whether some deficits could be reversed. This latter study made use of an already existing and safe clinical compound, oxytocin, to ameliorate impaired behaviors. But the choice of oxytocin arose from the specific deficits we observed (in which oxytocin has a very direct role).

As noted by the reviewer and as expected from the role of *Shank3* at the postsynapse, other postsynaptic components are likely to be affected by *Shank3*- deficiency. Indeed, findings from *Shank3*-mouse models support this notion and show changes in the levels of other PSD scaffolding components, subunits of glutamatergic receptors, as well as deficits in the synaptic actin cytoskeleton remodeling system, which is well associated with synaptic plasticity. Amongst the most reliable and reproducible findings from *Shank3* mouse models is the decrease in the levels of Homer, a scaffolding protein that interacts with *Shank3* at the PSD. In the *Shank3*-deficient rat model, we also found a significant decrease in Homer, in KO rat PFC samples, which validates the mouse models findings. We have added this finding to the Results section (subsection “Production and developmental phenotyping of the *Shank3*-deficient rat model”, first paragraph), Discussion, and to Figure 1—figure supplement 1 legend.

We have recently obtained preliminary data (not discussed in this paper) suggesting changes in the actin cytoskeleton components, similar to those reported in the *Shank3*-mouse models. While valuable, these findings are not informative by themselves and require further investigation to examine how such baseline changescontribute to the observed synaptic plasticitydeficits, so to be able to infer the direct relationship between both. This will be done by studying the dynamics of these and other synaptic components before and after in-vitro or in-vivo synaptic stimulation, and which is beyond the scope if the current study.

*5) The recovery of behavioral impairments by i.c.v. oxytocin is an exciting finding. However, the authors have not addressed which receptor type, expressed in which brain region(s), is responsible. Oxytocin can activate both vasopressin and oxytocin receptors, and these receptors are widely expressed throughout the brain. Since the authors have shown that the synaptic plasticity in the hippocampus is impaired whereas it remains intact in the PFC, oxytocin is presumably acting on the hippocampus to recover the attentional deficits and memory. Is there any experimental evidence to support this?*

We agree with the reviewer that oxytocin can activate both oxytocin and vasopressin receptors, which are widely spread across different brain regions and play and essential role in translating the oxytocin signal into cellular changes. We believe this question is important mainly because by discovering which receptors are activated by oxytocin and which downstream pathways are stimulated by this activation, we will be able to uncover new therapeutic targets for treatments. While highly important and significant to the field, we feel that addressing this question is best done in an independent study.

Regarding the second point raised by the reviewer, we respectfully disagree with the statement that “the synaptic plasticity in the hippocampus is impaired whereas it remains intact in the PFC”. PFC plasticity is not intact, as we clearly show that LTP in the PFC was reduced in both *Shank3*-Het and KO rats. Having that explained, one can’t conclude based on current available data as to which brain region is targeted and most benefited by the oxytocin administration. PFC would certainly be the more reasonable target of attention restoration by oxytocin, yet this has to be experimentally examined. As we stated for the first point, and as the reviewer implied, identifying the target receptor activated by oxytocin and the brain region targeted will help address this issue. In this regard, brain recording as well as imaging studies (e.g., fMRI) following oxytocin administrations can also be valuable tool to be considered. Yet as we noted above, while important and significant to the field, understanding the mechanisms by which oxytocin exerts its effect is, in our opinion, best left to an independent study.

*Again, social memory and attention while sometimes disrupted in ASD are not core features of ASD. The hippocampal deficits that are suggested here speak more to learning and memory and therefore, ID, in humans.*

This comment has been addressed at length in response to the reviewer’s general overview and in our response to his first and third comments.

[Editors’ note: the author responses to the re-review follow.]

*[…] This manuscript reports a valuable description of a Shank3 mutant rat, its cognitive deficits and rescue of the deficits tested by ICV oxytocin.*

*The authors are asked to address the sample sizes one last time. The authors say that the samples were largely constrained by the 25-50-25 inheritance pattern of WT-Hets-KOs. Yet the samples don't show this 25-50-25 distribution. They are very close to 1:1:1 overall. In the early tests – where mortality does not figure and where the distribution should be closer to Mendelian – the rats are at 9-15-12 which is a ratio of 0.6-1-0.8. However, in the core social memory tests (conducted later) the distributions are 12-13-9, 22-20-11, 9-17-14, 31-37-25, averaging 0.9-1-0.7. Then in the oxytocin rescue experiments (which are if anything more time consuming and subject to loss), the ratio is again very close to 1:1:1.*

*These ratios close to one are not easy to understand and are not explained by the authors' letter. The reason that this is important is because of the need to set sample size targets for behavioral tests prior to conducting the tests in order to avoid unconscious bias to creep into the experiment.*

*It might be of help to discuss how many animals in each pool and of each type were lost to surgery or some other fatality.*

*Was the person conducting the e-phys blind to the rat's genotype? And the person that analyzed the e-phys’ results?*

We thank the reviewer for raising these important questions, and we have divided our response into several sections.

Sample size:

We would like to begin by emphasizing that our minimum sample sizes for each experiment, chosen based on power analyses, were set before the experiments started, and that there were no intermediate analyses based on partial samples (and hence no alteration in sample sizes as the study progressed). Hence we do not think that there could be a source of unconscious bias that led to altered results.

We also want to apologize for the confusion that we introduced by the description of 1:2:1 ratios of WT, Hets, and KOs. This was meant to underscore that the breeding scheme, together with the focus on males, had impact on sample size because only one out of every eight pups was expected to be homozygous and male. In practice, we targeted a 1:1:1 ratio in these studies.

Mendelian ratios:

Based on this comment regarding Mendelian ratios, we have now reviewed a large series of serial litters (n=130 litters). We observed that there is a reduction in the number of homozygous knockout animals at weaning (292:555:200, corresponding to ratios of 0.53:1:0.36). It is for this reason that there are sometimes fewer animals in the KO group. This is now noted in the text.

Animal loss:

We do not think that animal loss played a biased role in the results. As we now note in the text, in the long-term SRM task, 1 WT and 2 KOs lost their caps, 1 Het and 1 KO died during surgery, and 1 Het was extremely stressed and thus was not subjected to oxytocin injection. For the 5-CSRT, 4 WT, 5 Het, and 3 KO rats were unable to reach performance criterion at 1 s and therefore were excluded from analyses. This is now noted in the text.

Binding of assessor:

Finally, we would like to also state that the electrophysiology experiments were done with the experimenter and data analyst blind to genotype. This is our standard approach for all studies and we had already noted this for some experiments but accidently omitted to note this for electrophysiology and in a few other places in the manuscript. We have now added this information to the Methods.